# A Pitfall in Conformal Prediction: When Shorter Intervals Are Not Better

## Abstract

Conformal prediction has become a cornerstone of distribution-free uncertainty quantification, conventionally evaluated by its coverage and interval length. This work critically examines the sufficiency of these standard metrics. We demonstrate that *the interval length might be deceptively improved through a counter-intuitive approach* termed Prejudicial Trick (PT), while the coverage remains valid. Specifically, for any given test sample, PT probabilistically returns an interval, which is either null or constructed using an adjusted confidence level, thereby preserving marginal coverage. While PT potentially yields a deceptively lower interval length, it introduces practical vulnerabilities: the same input can yield completely different prediction intervals across repeated runs of the algorithm. We formally derive the conditions under which PT achieves these misleading improvements and provide extensive empirical evidence across various regression and classification tasks. Furthermore, we introduce a new metric *interval stability* which helps detect whether a new conformal prediction method implicitly improves the length based on such PT-like techniques.

## 1 Introduction

Machine learning is rapidly evolving and has been successfully applied in numerous fields (Voulodimos et al., 2018; Brown et al., 2020). However, machine learning models, particularly deep learning models, often suffer from overconfidence issues (Guo et al., 2017; Minderer et al., 2021), making them unreliable for deployment in high-stakes areas such as medicine and finance (De Prado, 2018). Therefore, it is crucial to develop techniques for uncertainty quantification and calibrate the original model to enhance the reliability of predictions (Sullivan, 2015; Minderer et al., 2021; Smith, 2024).

Among all the uncertainty quantification methods, conformal prediction stands out due to its simplicity and distribution-free characteristics (Vovk et al., 2005; Shafer & Vovk, 2008; Angelopoulos & Bates, 2021). Conformal prediction is a post hoc approach for constructing prediction intervals, based on a non-conformity score calculated on a hold-out calibration set (Algorithm 2). Conformal prediction and its variants have demonstrated promising performances in numerous applications (Lei & Candès, 2021; Angelopoulos et al., 2022).

Generally, researchers evaluate the intervals returned by conformal prediction via two criteria: *coverage* and *interval length*. Firstly, a valid coverage ensures that the actual response value has a high probability of falling within the interval. Secondly, the interval is encouraged to be as short as possible, as a shorter interval provides more precise information about prediction uncertainties. These two evaluation metrics are commonly used in the literature (Tibshirani et al., 2019; Teng et al., 2022; Angelopoulos et al., 2023; He & Lam, 2024) and a branch of works improves the length with several different meaningful approaches (Romano et al., 2019; Izbicki et al., 2020; Teng et al., 2022; Guan, 2023; Stutz et al., 2022). This raises a crucial question about the potential pitfalls of focusing narrowly on these standard metrics: If a new algorithm outperforms existing ones on coverage and length, should it automatically be considered superior for practical deployment? Specifically,

---
**The key question:**

Can a conformal prediction method maintain valid coverage and *deceptively improve interval length metrics* through counter-intuitive constructions, while *introducing practical risks*?

---

Consider the following Example 1:

---

**Algorithm 1** Prejudicial Trick (PT)

---

1: **Input:** conformal prediction algorithm (base) $\mathcal{A}_{1-\alpha}(\cdot; \hat{\mu})$, test point $\boldsymbol{x}'$, probability $p$.
2: Generate a uniform random variable $U \sim \mathrm{Unif}([0, 1])$;
3: **if** $U > p$: **then**
4:     Interval $\mathcal{C}_{1-\alpha}(\boldsymbol{x}') = \hat{\mu}(\boldsymbol{x}')$ (regression tasks) or $\mathcal{C}_{1-\alpha}(\boldsymbol{x}') = \varnothing$ (classification tasks);
5: **else**
6:     Calculate the adjusted miscoverage rate $\alpha' = 1 - \frac{1-\alpha}{p}$;
7:     Interval $\mathcal{C}_{1-\alpha}(\boldsymbol{x}') = \mathcal{A}_{1-\alpha'}(\boldsymbol{x}'; \hat{\mu})$;
8: **end if**
9: **Output:** Interval $\mathcal{C}_{1-\alpha}(\boldsymbol{x}')$.

---

**Example 1** (The Pitfalls of Length.)**.** Two doctors, Alice and Bob, are estimating recovery time for patients after treatment. Conformal prediction with historical data reveals that $60\%$ of patients recover within $4$ years, and $80\%$ within $5$ years. When a new patient asks for an estimated recovery time, Alice and Bob adopt distinct strategies:
- Alice: Assign recovery time interval $[0, 4]$ years consistently;
- Bob: Assign recovery time interval $[0, 5]$ with probability $0.75$, while $[0, 0]$ with probability $0.25$.

For both strategies in Example 1, $60\%$ of patients fall in the estimated interval in expectation, thus satisfying the criteria for valid marginal coverage. Besides, Bob's approach yields a shorter average interval length $5 \times 75\% = 3.75 \ll 4$. Overall, Bob achieves a shorter interval while achieving the same coverage as Alice. However, Bob's strategy is flawed in its practical application, since (a) from the micro-level, Bob provides different intervals for the same patient if queried multiple times, and (b) from the macro-level, Bob randomly informs $25\%$ of patients that they will recover immediately after treatment regardless of their actual condition. The example is illustrated in Figure 3.

In this paper, inspired by the motivating example (Example 1), the *Prejudicial Trick* emerges as a practically invalid method that artificially shortens prediction intervals in conformal prediction (see Algorithm 1 and Figure 4). Instead of providing consistent intervals, PT assigns null intervals with a fixed probability to any test sample, and assigns confidence intervals with lower miscoverage rates in other cases to maintain the marginal coverage. While PT preserves marginal coverage and potentially reduces the average interval length, its rationale is less sound compared to standard methods like Vanilla Conformal Prediction (VCP). Specifically, PT suffers from two limitations:
- *Instability issues*: Repeated runs of PT produce different intervals for the same input;
- *Unfairness issues*: PT provides informative predictions for only a subset of test samples, while assigning uninformative null intervals to the rest[1].

From the theoretical perspective, we offer several theoretical results to provide deep understandings of PT regarding both coverage and length, and informally summarize them in Theorem 2.

**Theorem 2** (Theorem Summary)**.** *We term* base *as the base conformal prediction algorithm, term* PT *as the base algorithm with PT, and omit mild assumptions for clarity. For coverage, it holds that:*
- *PT satisfies marginal coverage guarantees under exchangeability assumptions (Theorem 4);*
- *PT guarantees conditional coverage if its base guarantees conditional coverage (Theorem 5);*
- *PT provably outperforms its base regarding the conditional coverage, under some conditions, even if the base does not satisfy the conditional coverage guarantees (Theorem 6).*

*For length, it holds that:*
- *PT achieves shorter average intervals than its base under some general conditions (Theorem 7);*
- *We provide sufficient conditions under which PT reduces the average interval length for both differentiable (Theorem 8) and non-differentiable (Theorem 11) length functions. Notably, these conditions are often satisfied in the common scenario of model misspecification (Remark 4).*
- *These results lead to corollaries for specific cases, such as when the length function is locally concave (Corollary 9) or when the base algorithm is VCP (Corollary 10).*
- *We also provide a failure case where PT cannot decrease the average length (Example 12).*

---

[1] In this paper, unfairness stems from the fact that a portion of samples are assigned null intervals in a run, even though each has an equal probability of being prejudiced. This differs from the unfairness concept grounded in conditional coverage (Zhao et al., 2020), where individuals are prejudiced based on their features.

From the experimental perspective, we verify our findings on various real-world datasets, regarding marginal and conditional coverage (Figure 1), and interval length (Table 2). Besides, we validate our findings under different settings, including different tasks (classification regimes in Table 4) and other conformal prediction algorithms (Conformalized Quantile Regression in Table 5).

However, the improvement on length is vacuous, as discussed in Section 3.5. To detect PT, We further introduce *Interval Stability*, a new metric to quantify the variation of the prediction for the same input over multiple runs. This metric is practically meaningful since it serves to identify and alert methods that implicitly or explicitly deploy invalid techniques like PT in the future (Remark 6).

**Remark 1.** Notably, we present PT not as a practical solution, despite its theoretical advantages on the coverage-length metric. Instead, it acts as a cautionary example that raises issues like instability and unfairness during deployment. PT represents the most direct way to *hack* the coverage-length metric. The fact that such a simple trick can succeed exposes a fundamental blind spot in the current evaluation paradigm. If standard metrics can be fooled this easily, they are likely vulnerable to more complex and subtle manipulations from increasingly sophisticated models.

## 2 RELATED WORK

Conformal prediction (Vovk et al., 2005) is mainly evaluated by the coverage-length metric. For coverage, conformal prediction provides finite sample guarantees under exchangeability assumptions (Vovk et al., 2005; Tibshirani et al., 2019; Barber et al., 2023), ensuring that prediction sets achieve the expected marginal coverage. Another related metric is conditional coverage, which is unachievable in finite sample settings without further assumptions (Vovk, 2012). Therefore, recent work focuses on various relaxations of conditional coverage (Barber et al., 2020; Gibbs et al., 2024).

Another metric is the average *interval length*, as shorter intervals are generally more informative (Lei et al., 2018; Sadinle et al., 2018). Numerous methods aim to construct adaptive intervals that reduce length while maintaining valid coverage. One line of work designs alternative non-conformity score functions: for example, Romano et al. (2019) integrate quantile regression, while Guan (2023) propose a localized method that adapts to test-time information. Other score functions have been proposed in (Feldman et al., 2021; Alaa et al., 2023; Han et al., 2022; Teng et al., 2022). Of particular interest is the method of Izbicki et al. (2020), which estimates the asymptotic conditional distribution of the non-conformity scores and constructs prediction intervals based on high-density regions. Another line of work involves a training procedure for interval-length optimization, such as CPL (Kiyani et al., 2024), CP-Gen (Bai et al., 2022), ConfTr (Stutz et al., 2022), and BoostedCP (Xie et al., 2024). Different from existing methods that provide principled and meaningful advances in conformal prediction, our proposed prejudicial trick achieves efficiency gains through an illusory construction, making the length improvement non-substantive.

Besides coverage and length, several auxiliary metrics have been introduced. These include *excess* and *deficit* (Seedat et al., 2023), which measures the extent to which the prediction intervals are unnecessarily wide or insufficiently narrow; *false positive rate* (Fisch et al., 2022), which improves precision by limiting the number of incorrect labels in classification settings; and *conditional weighted coverage* (Jensen et al., 2024)—a hybrid metric that takes both coverage and length into regard. Among them, a particularly important evaluation criterion is *group coverage* (Cauchois et al., 2021), which assesses the coverage and interval length across population subgroups defined by features or response magnitudes. However, in practice, people still tend to prioritize the coverage and length metrics (Lei et al., 2017; Cresswell et al., 2024; Zhang et al., 2024; Xu et al., 2025). Notably, this paper introduces a new metric distinct from existing approaches. Unlike existing studies that design metrics mainly to showcase favorable performance but often struggle with length, we show that PT potentially surpasses its base model in length while performing poorly under the new metric. We provide additional related works on conformal prediction and interval regression in Appendix H.

## 3 PREJUDICIAL TRICK WITH DECEPTIVE IMPROVEMENT

In this section, we challenge the coverage-length metric in conformal prediction by constructing a trick in Section 3.2. Specifically, this trick potentially improves the interval length while maintaining the coverage, yet it introduces instability and unfairness issues. We further investigate how this trick influences the coverage (Section 3.3) and the length (Section 3.4) theoretically and empirically. Finally, we discuss more details on the deceptive improvements of this trick in Section 3.5.

Table 1: Results for the synthetic datasets (motivating example in Section 3.2). Comparison between VCP and PT-VCP under different $\alpha$ levels, regarding length and coverage.

| $\alpha$ | $p$ | VCP | | PT-VCP | |
|---|---|---|---|---|---|
| | | Coverage | Length | Coverage | Length |
| 0.10 | 0.96 | $0.906 \pm 0.004$ | $22.894 \pm 0.138$ | $0.909 \pm 0.005$ | $\mathbf{22.614} \pm 0.254$ |
| | 0.98 | $0.906 \pm 0.004$ | $22.894 \pm 0.138$ | $0.904 \pm 0.004$ | $\mathbf{22.714} \pm 0.165$ |
| 0.20 | 0.96 | $0.792 \pm 0.011$ | $21.886 \pm 0.125$ | $0.799 \pm 0.011$ | $\mathbf{21.255} \pm 0.136$ |
| | 0.98 | $0.792 \pm 0.011$ | $21.886 \pm 0.125$ | $0.796 \pm 0.010$ | $\mathbf{21.589} \pm 0.149$ |

## 3.1 PRELIMINARY

**Conformal Prediction.** Conformal prediction creates statistically rigorous uncertainty sets for any predictive model. Given $X$ as the input and $\alpha \in (0, 1)$ as the miscoverage rate, conformal prediction returns an uncertainty set[2] $\mathcal{C}_{1-\alpha}(X)$ that satisfies

$$\mathbb{P}(y \in \mathcal{C}_{1-\alpha}(X)) \geq 1 - \alpha, \tag{1}$$

where $y$ denotes the true response of feature $X$. We omit the detailed discussion in Appendix C.

**Notations.** Let $\{Z_i\}_{i=1}^n$ denote $n$ i.i.d. samples drawn from the distribution $\mathcal{P}_Z$, where $Z \in \mathbb{R}$. Denote $\{Z_{(i)}\}_{i=1}^n$ as the order statistics of $\{Z_i\}_{i=1}^n$ arranged in decreasing order, *i.e.*, $Z_{(1)} \geq Z_{(2)} \geq \cdots \geq Z_{(n)}$. The empirical $\tau$-th quantile with $n$ samples is defined as $\hat{Q}_\tau(\{Z_i\}_{i=1}^n) := Z_{(\lceil (n+1)(1-\tau) \rceil)}$. Let $\varnothing$ denote the empty set, and $\mathbb{I}(\cdot)$ denote the indicator function. For a given set $\mathcal{C}$, let $|\mathcal{C}|$ denote the measure of the set. In this paper, we denote a trained conformal prediction algorithm that directly outputs the $1 - \alpha$ confidence interval given a test point as $\mathcal{A}_{1-\alpha}(\cdot; \hat{\mu})$ for simplicity, where $\hat{\mu}$ is the machine learning algorithm used in the conformal prediction algorithm.

## 3.2 PREJUDICIAL TRICK

In this section, we propose a trick (Algorithm 1) used in conformal prediction with the intuition from Example 1 and deploy this trick in a motivating example on a synthetic dataset. The experiment results in Table 1 demonstrate that this trick improves the interval length while maintaining the marginal coverage compared to its base. We begin with the construction process of this trick:

**Construction Process.** For each test point, assign a null set[3] with probability $1 - p$, and assign the interval with an adjusted miscoverage rate $\alpha' = 1 - \frac{1-\alpha}{p}$ for the remaining $p$ portion of test points, where $p \in (1 - \alpha, 1)$. Overall, for a new test point $x'$, the interval is constructed as Equation (2):

$$\mathcal{C}_{1-\alpha}^{\text{PT}}(x') = \begin{cases} \text{null set} & \text{with the probability } 1 - p, \\ \mathcal{C}_{1-\alpha'}^{\text{CP}}(x') & \text{with the probability } p. \end{cases} \tag{2}$$

We call this process Prejudicial Trick (PT, see in Algorithm 1). Note that PT can be directly applied to any base conformal prediction algorithm. To illustrate that PT improves length without sacrificing marginal coverage, we empirically consider a motivating example in Example 3

**Example 3** (Synthetic Dataset). Consider a regression setting where the true underlying model is a linear model with the Gaussian mixture noise, given by $Y = X^\top \beta + \epsilon$, with $X \sim \mathcal{N}(\mathbf{0}, I_2)$. The noise term $\epsilon$ follows $\mathcal{N}(\mu, 1)$ with probability 0.5, and $\mathcal{N}(-\mu, 1)$ with probability 0.5. The training fold, calibration fold, and test fold are generated based on this underlying distribution. We deploy VCP (Algorithm 2) and PT-VCP under such regimes. We refer to Appenix G.1.1 for more details.

**Results and Discussions of Example 3.** Tabel 1 illustrates the results of Example 3, where *PT-VCP improves the length while maintaining the marginal coverage compared to VCP*. The results validate that the length-coverage metric could be hacked by invalid tricks like PT. The insights are as follows:

---

[2]Conformal prediction either returns a set measured by its size (for classification tasks), or returns an interval measured by its length (for regression tasks). We do not distinguish between these terms throughout the paper.

[3]The null set represents a set of measure zero. It can be an empty set, or a single-point set in regression.

the construction of $\epsilon$ guarantees $\mathcal{C}^{\mathrm{CP}}_{1-\alpha'}$ close to $\mathcal{C}^{\mathrm{CP}}_{1-\alpha}$ regarding the length when choosing proper $\alpha$ and $p$. Therefore, PT potentially improves the average length when averaging with those null sets.

Notably, Barber et al. (2020) propose a method similar to PT. Unlike PT which emphasizes the potential length improvement, Barber et al. (2020) mainly center on conditional coverage metrics. Moreover, we extend the scope of PT in Remark 2 by relaxing the notion of the null set.

**Remark 2** (The extension of PT). PT in Algorithm 1 heavily relies on the notion of null sets. Fortunately, one can extend this null set with an interval returned by conformal prediction with a small coverage rate. For example, PT can be constructed as follows:

$$\mathcal{C}^{\mathrm{PT}}_{1-\alpha}(\boldsymbol{x}') = \begin{cases} \mathcal{C}^{\mathrm{CP}}_{1-\alpha'_1}(\boldsymbol{x}') & \text{with the probability } 1-p, \\ \mathcal{C}^{\mathrm{CP}}_{1-\alpha'_2}(\boldsymbol{x}') & \text{with the probability } p, \end{cases} \tag{3}$$

where $(1-p)\alpha'_1 + p\alpha'_2 = \alpha$ which guarantees the marginal coverage and $\alpha'_1$ is sufficiently small. We mainly consider the null set in this paper to simplify the related discussions.

### 3.3 COVERAGE

This section investigates the coverage guarantee of conformal prediction with PT. We prove that PT maintains the marginal coverage guarantees (Theorem 4) and conditional coverage guarantees (Theorem 5, Theorem 6). The empirical validation in Figure 1 supports the theoretical findings.

**Marginal Coverage.** Theorem 4 proves that PT maintains the valid marginal coverage guarantees.

**Theorem 4** (Marginal Coverage Guarantee). *Assume that the exchangeability assumption holds (see Proposition 14 for more details). Then the interval returned by Algorithm 1 with the adjusted miscoverage rate $\alpha' = 1 - \frac{1-\alpha}{p}$ guarantees that*

$$\mathbb{P}(y' \in \mathcal{C}^{PT}_{1-\alpha}(\boldsymbol{X}')) \geq 1 - \alpha, \tag{4}$$

*where $(\boldsymbol{X}', y')$ denotes a new test point.*

The intuition behind Theorem 4: the null set (with probability $1-p$) and the enlarged interval set with miscoverage $\alpha'$ (with probability $p$) reach the marginal coverage guarantees $p(1-\alpha') = 1-\alpha$.

**Conditional Coverage.** Theorem 5 further investigates the effect of PT on the condition coverage. Specifically, the interval returned by PT $\mathcal{C}^{\mathrm{PT}}_{1-\alpha}(\boldsymbol{X}')$ keeps the conditional coverage guarantees if the interval returned by its base algorithm $\mathcal{C}^{\mathrm{CP}}_{1-\alpha}(\boldsymbol{X}')$ satisfies the conditional guarantees.

**Theorem 5** (Conditional Coverage Guarantee). *Let $\mathcal{C}^{CP}_{1-\alpha}(\boldsymbol{X}')$ and $\mathcal{C}^{PT}_{1-\alpha}(\boldsymbol{X}')$ denote the returned interval of $\boldsymbol{X}'$ respectively. If for any $\alpha$, $\mathbb{P}(y \in \mathcal{C}^{CP}_{1-\alpha}(\boldsymbol{X}') \mid \boldsymbol{X}') \geq 1 - \alpha$ holds for $\boldsymbol{X}'$ almost surely, then $\mathbb{P}(y \in \mathcal{C}^{PT}_{1-\alpha}(\boldsymbol{X}') \mid \boldsymbol{X}') \geq 1 - \alpha$ holds for any $\alpha$ and for $\boldsymbol{X}'$ almost surely as well.*

The key insight behind Theorem 5: The randomness within PT is independent of the specific input. Therefore, such randomness is averaged out given a specific input, thus keeping the conditional coverage unchanged. Specifically, the conditional coverage is calculated as $p(1-\alpha') = 1-\alpha$.

**Remark 3** (Comparison to the Tradeoffs between Conditional Coverage and Interval Length). Existing works on conformal prediction have analyzed the potential tradeoffs between conditional coverage and interval length (Barber et al., 2020; Gibbs et al., 2024). Our work differs from this line, since PT does not operate by creating such a trade-off. Specifically, Theorem 5 validates that PT does not violate the conditional coverage. We will provide a further discussion in Section 3.5.

Theorem 5 requires that the base algorithm satisfies the conditional coverage guarantees. However, this requirement does not always hold in practice. We next prove in Theorem 6 that PT still exhibits the potential to outperform the base algorithm even when the requirement does not hold in practice.

**Theorem 6** (Sufficient Condition of Conditional Coverage Guarantees). *Let $f_{\mathcal{A}}(\alpha)$ denote the true conditional miscoverage rate within the subset $\mathcal{A}$, namely, $\mathbb{P}(y \in \mathcal{C}^{CP}_{1-\alpha}(\boldsymbol{X}) \mid \boldsymbol{X} \in \mathcal{A}) = 1 - f_{\mathcal{A}}(\alpha)$ where $\mathcal{C}^{CP}_{1-\alpha}(\boldsymbol{X})$ denotes the prediction set returned by the base algorithm. Define $\mathcal{F}(p) = p f_{\mathcal{A}}(1 - (1-\alpha)/p)$ where $p \in (1-\alpha, 1)$ represents the parameter in Algorithm 1. If $\mathcal{F}(\cdot)$ satisfies that $\mathcal{F}(1) - \mathcal{F}(p) \geq 1 - p$, then it holds that*

$$\mathbb{P}(y \in \mathcal{C}^{PT}_{1-\alpha}(\boldsymbol{X}) \mid \boldsymbol{X} \in \mathcal{A}) \geq \mathbb{P}(y \in \mathcal{C}^{CP}_{1-\alpha}(\boldsymbol{X}) \mid \boldsymbol{X} \in \mathcal{A}), \tag{5}$$

*where $\mathcal{C}^{CP}_{1-\alpha}(\boldsymbol{X})$ denotes the prediction set returned by PT.*

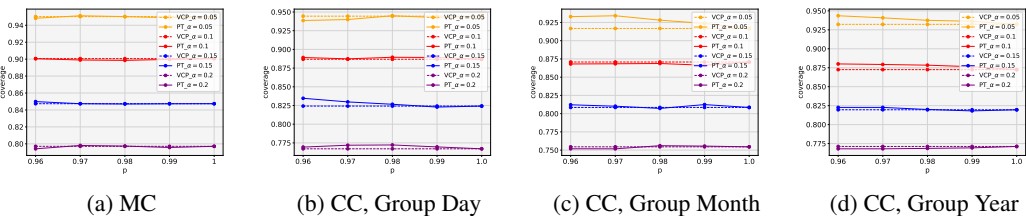

Figure 1: Comparing the (a) marginal coverage and (b, c, d) conditional coverage between VCP with and without PT. Results demonstrate that (1) PT would not significantly change the marginal coverage; (2) PT has better conditional coverage compared to the base algorithm (Theorem 6).

The key insight behind Theorem 6: The conditional miscoverage rate of PT arises from two probabilistic components. For the null set component, the miscoverage rate exceeds that of the base algorithm; for the other component, the miscoverage rate is lower. Therefore, by averaging these two components, PT potentially outperforms its base algorithm in terms of conditional coverage under certain conditions on the miscoverage function $f_{\mathcal{A}}(\alpha)$. Notably, Theorem 6 contains results on both conditional coverages (for one-point set $\mathcal{A}$) and group coverages (for regular set $\mathcal{A}$).

**Experiments.** We compare marginal and conditional coverage rates returned with and without PT on BIKE dataset (Fanaee-T, 2013) using VCP (Algorithm 2) as the base algorithm. We omit the implementation details here and refer to Appendix G.2.2 for details. The results in Figure 1 demonstrate that (a) PT preserves the marginal coverage (Figure 1a); (b) When VCP fails to guarantee the group coverage, PT-VCP fails as well. Experimental results demonstrate that PT achieves comparable group coverage with its base models (Figure 1b, Figure 1c, Figure 1d). We provide more experiments on group coverage with different real-world datasets in Appendix F.1.

### 3.4 LENGTH

In this section, we investigate the sufficient conditions under which PT improves interval length while keeping the coverage unchanged, and further conduct experiments to validate the theoretical findings. We first propose Theorem 7 as a weak sufficient condition. We then derive a more informative condition under both differentiable regimes (Theorem 8) and non-differentiable regimes (Theorem 11). We further discuss the special cases on the local concave assumption (Corollary 9) and VCP regimes (Corollary 10), and find that misspecification usually satisfies the sufficient conditions (Remark 4). We finally present a failure case in Example 12 when PT cannot outperform its base regarding the interval length.

Experiment results on various datasets in Table 2 align closely with the theoretical results. Besides, we conduct experiments on classification tasks (Table 4), different base algorithms (Table 5), and conduct ablation studies on different hyperparameters (Figure 6-Figure 15).

**Additional Notations.** We introduce the following notations to facilitate the discussions in this section. Let $\alpha$ denote the miscoverage rate and $s(\boldsymbol{x}, y; \hat{\mu})$ denote the score function, where $\hat{\mu}(\cdot)$ denotes the learned model. Let $\mathcal{C}_{1-\alpha}^{\mathrm{CP}}(\boldsymbol{x})$ denote the interval returned by conformal prediction at point $\boldsymbol{x}$, and $\mathcal{C}_{1-\alpha}^{\mathrm{PT}}(\boldsymbol{x})$ denote the interval returned by its PT-variant (Algorithm 1). Let $\mathcal{L}(\boldsymbol{x}, 1-\alpha; s)$ denote the length of the returned interval at point $\boldsymbol{x}$ with miscoverage $\alpha$, i.e., $|\mathcal{C}_{1-\alpha}^{\mathrm{CP}}(\boldsymbol{x})|$.

We next prove a series of sufficient conditions under which PT improves the interval length, starting from Theorem 7 which provides a simple and straightforward sufficient condition.

**Theorem 7** (General Sufficient Condition). *If $\mathcal{L}$ satisfies the following condition:*

$$\exists p \in (1-\tilde{\alpha}, 1) \quad s.t. \quad p\mathbb{E}\left(\mathcal{L}\left(\boldsymbol{x}, \frac{1-\tilde{\alpha}}{p}; s\right)\right) < \mathbb{E}(\mathcal{L}(\boldsymbol{x}, 1-\tilde{\alpha}; s)), \quad (6)$$

*where $\tilde{\alpha}$ denotes the miscoverage rate and the expectation is taken over $\boldsymbol{x}$. Then the interval length returned by PT (with parameter $p$) outperforms that of its base algorithm, namely*

$$\mathbb{E}|\mathcal{C}_{1-\tilde{\alpha}}^{PT}(\boldsymbol{X}')| < \mathbb{E}|\mathcal{C}_{1-\tilde{\alpha}}^{CP}(\boldsymbol{X}')|, \quad (7)$$

*where the expectation is taken over the testing point $\boldsymbol{X}'$.*

The intuition behind Theorem 7 is pretty simple: PT assigns null sets with a fixed probability whose measure is zero, thus potentially reducing the average length. Although the sufficient condition in Theorem 7 is general, the absence of additional assumptions makes it uninformative in practice. To obtain more insights, we next introduce a differentiable assumption in Theorem 8.

**Theorem 8** (First-order Condition). *Assume $\mathcal{L}$ is first-order differentiable and satisfies*

$$\mathbb{E}\left(\frac{\mathcal{L}(\boldsymbol{x}, 1-\tilde{\alpha}; s)}{1-\tilde{\alpha}}\right) > \mathbb{E}\left(\frac{\partial}{\partial\alpha}\mathcal{L}(\boldsymbol{x}, \alpha; s)\bigg|_{\alpha=1-\tilde{\alpha}}\right), \tag{8}$$

*where $\tilde{\alpha}$ denotes the miscoverage rate and the expectation is taken over $\boldsymbol{x}$. Then there exists a parameter $p$ in Algorithm 1, such that the interval length returned by PT outperforms that of its base algorithm, namely*

$$\mathbb{E}|\mathcal{C}_{1-\tilde{\alpha}}^{PT}(\boldsymbol{X}')| < \mathbb{E}|\mathcal{C}_{1-\tilde{\alpha}}^{CP}(\boldsymbol{X}')|, \tag{9}$$

*where the expectation is taken over the testing point $\boldsymbol{X}'$.*

Theorem 8 follows the insights of Theorem 7, and further utilizes Equation 8 as the sufficient condition, which characterizes the local behavior of the interval length function. Theorem 8 provides more insights on when and how PT outperforms its base algorithm regarding the length metrics. We next derive a localized concave condition in Corollary 9 based on Theorem 8.

**Corollary 9** (Localized Concave Conditions). *Under the settings in Theorem 8, the sufficient condition in Equation 8 holds if $\mathbb{E}(\mathcal{L}(\boldsymbol{x}, \alpha; s))$ is locally concave on $\alpha \in [0, 1-\tilde{\alpha}]$.*

Corollary 9 provides a condition under which PT outperforms its base algorithm regarding length. Consider a regression problem with additive noise $y = f^*(x) + \epsilon$. If the noise distribution exhibits local concavity and the base model approximates the true function $f^*$ well, then the length function $\mathbb{E}(\mathcal{L}(\boldsymbol{x}, \alpha; s))$ generally satisfies the localized concavity property. Consequently, the performance improvement of PT is guaranteed by Corollary 9. This inspires the construction of Example 3.

Besides, Corollary 10 focuses on the settings of deploying VCP. Since VCP returns the same length for each individual, the expectation operator in Theorem 8 degenerates.

**Corollary 10** (Deterministic Case). *Under the settings in Theorem 8, if applying VCP (Algorithm 2) as the base algorithm, the sufficient condition in Equation 8 holds if*

$$\frac{\mathcal{L}(\boldsymbol{x}, 1-\tilde{\alpha}; s)}{1-\tilde{\alpha}} > \frac{\partial}{\partial\alpha}\mathcal{L}(\boldsymbol{x}, \alpha; s)\bigg|_{\alpha=1-\tilde{\alpha}}, \tag{10}$$

*where the expectation operator degenerates due to the characteristics of VCP.*

Unfortunately, real-world applications may not satisfy the differentiability assumption in Theorem 8. Therefore, we relax this assumption and derive a secant sufficient condition in Theorem 11.

**Theorem 11** (Secant Sufficient Condition). *If there exists $u \in (1-\tilde{\alpha}, 1)$, such that*

$$\frac{\mathbb{E}(\mathcal{L}(\boldsymbol{x}, 1-\tilde{\alpha}; s))}{1-\tilde{\alpha}} > \frac{\mathbb{E}(\mathcal{L}(\boldsymbol{x}, u; s)) - \mathbb{E}(\mathcal{L}(\boldsymbol{x}, 1-\tilde{\alpha}; s))}{u - (1-\tilde{\alpha})}, \tag{11}$$

*where $\tilde{\alpha}$ denotes the miscoverage rate and the expectation is taken over $\boldsymbol{x}$. Then there exists a parameter $p = (1-\tilde{\alpha})/u$ in Algorithm 1, such that the interval length returned by PT outperforms its base algorithm, namely,*

$$\mathbb{E}|\mathcal{C}_{1-\tilde{\alpha}}^{PT}(\boldsymbol{X}')| < \mathbb{E}|\mathcal{C}_{1-\tilde{\alpha}}^{CP}(\boldsymbol{X}')|, \tag{12}$$

*where the expectation is taken over the testing point $\boldsymbol{X}'$.*

Theorem 11 shares similar intuitions with Theorem 8, and further relaxes the differentiability assumption by comparing the secant slopes. Informally, PT achieves smaller average lengths than its base algorithm when the length function does not grow extremely fast within the region $(1-\tilde{\alpha}, 1)$.

**Remark 4** (Relationship Between Misspecification and Sufficient Condition). Model misspecification is a common practical scenario that aligns with our theoretical analysis, as it typically satisfies the sufficient conditions in Theorem 8 and Theorem 11 (Wang & Blei, 2020; Huang et al., 2023). Specifically, misspecification leads to a residual with a non-zero mean, resulting in a non-convex length function. This outcome is closely related to the local concavity condition in Corollary 9. For this reason, we employ misspecification regimes in most of our experiments.

Table 2: Comparison of performance between VCP and PT-VCP in regression tasks across different datasets ($\alpha = 0.1$).

| METHOD | | VCP | | PT-VCP | |
|---|---|---|---|---|---|
| DATASET | BIAS | COVERAGE | LENGTH | COVERAGE | LENGTH |
| MEPS-19 | 20 | 0.90 $\pm0.000$ | 42.34 $\pm0.228$ | 0.90 $\pm0.000$ | **41.92** $\pm0.389$ |
| MEPS-20 | 20 | 0.90 $\pm0.000$ | 41.98 $\pm0.116$ | 0.90 $\pm0.000$ | **41.41** $\pm0.241$ |
| MEPS-21 | 20 | 0.90 $\pm0.004$ | 42.28 $\pm0.112$ | 0.90 $\pm0.000$ | **41.90** $\pm0.300$ |
| BIKE | 10 | 0.90 $\pm0.000$ | 20.46 $\pm0.018$ | 0.90 $\pm0.004$ | **19.59** $\pm0.018$ |
| BLOG-DATA | 20 | 0.90 $\pm0.004$ | 41.67 $\pm0.336$ | 0.90 $\pm0.000$ | **41.13** $\pm0.416$ |
| BIO | 10 | 0.90 $\pm0.004$ | 21.13 $\pm0.336$ | 0.90 $\pm0.000$ | **20.44** $\pm0.031$ |
| FACEBOOK-1 | 10 | 0.90 $\pm0.000$ | 20.81 $\pm0.036$ | 0.90 $\pm0.000$ | **20.80** $\pm0.179$ |
| FACEBOOK-2 | 10 | 0.90 $\pm0.000$ | **20.97** $\pm0.067$ | 0.90 $\pm0.000$ | 21.01 $\pm0.179$ |
| CONCRETE | 5 | 0.90 $\pm0.013$ | 10.32 $\pm0.009$ | 0.89 $\pm0.009$ | **9.87** $\pm0.031$ |
| STAR | 5 | 0.91 $\pm0.004$ | 10.14 $\pm0.004$ | 0.91 $\pm0.004$ | **9.63** $\pm0.027$ |

**Remark 5** (Extensions of PT Beyond Conformal Prediction). While Theorem 7 can be extended to other interval estimation tasks, the relevance between PT and conformal prediction lies in the model misspecification (Remark 4). Specifically, model misspecification serves as a potential sufficient condition where PT works, and it generally appears in real-world applications of conformal prediction. However, the existence of model misspecification does not always hold for other interval estimation tasks, and therefore limits the extensions of PT beyond conformal prediction.

However, the aforementioned sufficient conditions are not always satisfied. We present a failure case in Example 12 and illustrate the empirical validation in Figure 5.

**Example 12** (Failure Case). *If the values of non-conformity score in VCP follows a Gaussian distribution over randomness in $\boldsymbol{x}$, then for all $\alpha \in (0, 1)$, and all $p \in (1 - \alpha, 1)$, it holds that*

$$\mathbb{E}|\mathcal{C}_{1-\alpha}^{PT}(\boldsymbol{X}')| > \mathbb{E}|\mathcal{C}_{1-\alpha}^{CP}(\boldsymbol{X}')|. \tag{13}$$

Example 12 demonstrates that PT does not always outperform its base algorithm regarding the length-coverage metric. However, our goal is not to present PT as a universally applicable method, but to demonstrate a fundamental flaw in the coverage-length evaluation. To achieve this, the existence of any realistic scenarios where PT can create deceptively shorter intervals is sufficient.

**Experiment.** We conduct several experiments comparing the length returned with and without PT using VCP (Algorithm 2) on MEPS19-21 (Cohen et al., 2009), BIKE (Fanaee-T, 2013), BLOG-DATA (Buza, 2014), BIO (Rana, 2013), FACEBOOK1-2 (Singh, 2015), CONCRETE (Yeh, 1998), STAR (Achilles et al., 2008). To simulate model misspecification, we manually add bias to the label (as shown in the bias column in Table 2). We refer to Appendix G.2 for setting details. The results in Table 2 demonstrate that PT generally achieves smaller average lengths compared to its base algorithm in most cases (9 out of 10). Besides, we conduct more experiments and ablations:

- We compare the length of VCP and PT-VCP under classification tasks in Table 4;
- We use CQR (Romano et al., 2019) as the base algorithm on regression tasks in Table 5;
- We conduct ablation studies on hyperparameter $p$ and misspecification level $\mu$ in Appendix F.2.

## 3.5 DECEPTIVE IMPROVEMENT

We prove that PT preserves (conditional) coverage guarantees in Section 3.3 and achieves shorter prediction intervals under certain conditions in Section 3.4. Despite these theoretical benefits, PT is poorly suited for practical deployment. The primary issue is that PT introduces randomness, causing prediction intervals to vary across different runs. This inherent instability undermines the method's reliability. Besides, the problem becomes more dramatic in the scenario of Remark 2, where the individuals are grouped with different miscoverage rates. This randomness makes it impossible for a user to identify their assigned group, making the confidence interval meaningless. Therefore, while PT may appear superior based on the traditional length-coverage metric, its practical instability makes it unsuitable for real-world deployment. This discrepancy challenges the sufficiency of the length-coverage metric itself, suggesting it is not a complete measure of a method's practical utility.

Table 3: Comparison between VCP and VCP with PT regarding interval stability.

| Method | meps-19 | meps-20 | meps-21 | bike | blog-data | bio | facebook-1 | facebook-2 | concrete | star |
|--------|---------|---------|---------|------|-----------|-----|------------|------------|----------|------|
| VCP | 0.00 ±0.000 | 0.00 ±0.000 | 0.00 ±0.000 | 0.00 ±0.000 | 0.00 ±0.000 | 0.00 ±0.000 | 0.00 ±0.000 | 0.00 ±0.000 | 0.00 ±0.000 | 0.00 ±0.000 |
| PT-VCP | 1.26 ±0.015 | 1.24 ±0.008 | 1.26 ±0.011 | 0.58 ±0.002 | 1.23 ±0.014 | 0.61 ±0.001 | 0.62 ±0.005 | 1.19 ±0.006 | 1.14 ±0.012 | 1.14 ±0.007 |

## 4 INTERVAL STABILITY

In this section, we propose *interval stability* (Definition 1) which measures the randomness in each run of conformal prediction. We begin with the definition of interval stability in Definition 1.

**Definition 1** (Interval Stability). *Let $X$ denote a data point with returned confidence interval $C_{1-\alpha}(X)$, and let $|\cdot|$ denote a certain measure of the interval (e.g., its length). Let $\mathcal{A}$ denote the conformal prediction algorithm, and $\mathcal{D}_{ca}$ the calibration dataset. The interval stability is defined as*

$$\mathbb{E}_X \left[ \mathrm{Var}_{\mathcal{A}|X,\mathcal{D}_{ca}} \left( |\mathcal{C}_{1-\alpha}(X)| \right) \right]. \tag{14}$$

The interval stability captures the expected variability of the interval size conditional on the test point and calibration randomness. Intuitively, it captures the inconsistency of the returned intervals when the algorithm is run multiple times on the same test point and calibration dataset.

Due to the stochastic nature of PT, it tends to produce a large interval stability, implying the practical instability issues. We prove in Proposition 13 that PT indeed introduces a non-zero interval stability.

**Proposition 13.** *Following the notations in Section 3.4, it holds that*

$$\mathbb{E}_X \left[ \mathrm{Var}_{\mathcal{A}|X,\mathcal{D}_{ca}} \left( |\mathcal{C}_{1-\tilde{\alpha}}^{PT}(X)| \right) \right] = p(1-p) \left( \mathbb{E} \left( \mathcal{L}(\boldsymbol{x}, (1-\tilde{\alpha})/p; s) \right) \right)^2 > 0. \tag{15}$$

Notably, the *interval stability* metric is not just for detecting our specific PT construction, but serves as a safeguard to ensure that future advancements in conformal prediction are genuine and reliable, rather than arising from the unprincipled randomness. As the community pushes for ever-shorter prediction intervals, there is a risk that increasingly complex methods might implicitly introduce forms of randomness that offer deceptive gains. We refer to Remark 6 for a detailed discussion.

**Remark 6** (Why Interval Stability). Interval stability is still meaningful even if existing approaches in conformal prediction do not always rely on the randomness. In the existing literature, numerous approaches claim a superior performance through a smaller interval length, under the traditional coverage-length metric. In this paper, we show that randomness may break this metric through PT due to practical issues. This raises concerns that as methods become increasingly complex, they may implicitly utilize similar randomness to improve the length. Such effects may be unintended but hard to be recognized. To address this issue, interval stability serves as a tool for detecting such issues and highlighting the risks inherent in the current reliance on the length-coverage metric alone.

**Experiment.** We follow the same setting as the experiment in Section 3.4, but use interval stability as the metric. Table 3 demonstrates that interval stability successfully detects the vacuous randomness in PT. We defer to Appendix F.1 for more experiments with CQR and classification regimes.

We acknowledge that for deterministic methods, interval stability will indeed be zero. This is by design. The metric is not intended to replace coverage and length, but to complement them, acting as a specific check against the kind of vacuous randomness we identify. A value of zero is a *pass* on this specific test, confirming the method's deterministic nature for a given input.

## 5 CONCLUSION

In this paper, we demonstrate a pitfall in how conformal prediction methods are evaluated. We introduce PT, a technique that hacks the conventional coverage-length metric by producing deceptively shorter intervals while preserving coverage guarantees. However, PT relies on the randomness that leads to instability: the algorithm can produce different prediction sets for a given input on different runs. This creates practical issues in real-world, high-stakes scenarios. Our theoretical and empirical results confirm that while PT appears superior, its foundation is flawed. This discrepancy challenges the completeness of the coverage-length metric. Consequently, we propose Interval Stability as a diagnostic tool, which helps flag the potential vacuous randomness for a newly proposed method.

ETHICS STATEMENT

This research focuses on the pitfall in conformal prediction. Our work does not involve human subjects. Therefore no Institutional Review Board (IRB) approval was required. All experiments were conducted on standard, publicly available benchmarks, which are widely used in the machine learning community. Our research does not involve the collection of new data, nor does it process personally identifiable or sensitive information, thus mitigating concerns related to data privacy and security.

REPRODUCIBILITY STATEMENT

To ensure the reproducibility of our work, we provide detailed descriptions of our theoretical results and experimental setup. The theoretical results presented in Section 3.3 and Section 3.4 are accompanied by complete mathematical proofs in Appendix D. Our full experimental setup is described in Appendix G. The source code is provided in the supplementary material and will be made publicly available upon publication.

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

# Appendix

We firstly restate our contributions in Appendix A. Then we present more discussions in Appendix B. In Appendix C, we introduce the conformal prediction and we provide missing proofs in Appendix D. In Appendix E, we exhibit the omitted illustrations in our paper. In Appendix F, we illustrate the omitted experimental results. In Appendix G, we present implementation details of our experiments. In Appendix H, we exhibit more related works. In Appendix I, we clarify the use of large language models in our paper.

## A  CONTRIBUTIONS RESTATEMENT

We summarize our contributions as follows:

- We observe that the traditional coverage-length criteria in conformal prediction might be hacked using a counter-intuitive method PT, (Algorithm 1), since PT might deceptively improve the length while maintaining the coverage but raises fairness issues;

- We theoretically derive in Theorem 7 the conditions under which PT deceptively improves length, while keeping valid marginal coverage (Theorem 4) and conditional coverage (Theorem 5, Theorem 6). We further derive several sufficient conditions under which PT improves length with first-order differentiability assumption (Theorem 8) or without first-order differentiability assumption (Theorem 11);

- We propose a new metric in Section 4, termed interval stability. Interval stability measures the variance of the prediction interval over the input introduced by the conformal prediction algorithms, helping to mitigate the adverse impacts of PT.

## B  MORE DISCUSSIONS

**Similarity between PT and method discussed in Barber et al. (2020).** In Section 3.2, we mention the similarity between our PT method and the randomness discussed in Barber et al. (2020). While the mechanism in our work bears a structural resemblance to that in Barber et al. (2020), our motivation and conclusion are fundamentally different. Barber et al. (2020) investigate the inherent trade-offs required to achieve conditional coverage, using randomization as a tool to explore theoretical limits. In contrast, our work focuses on the evaluation paradigm itself. We use PT not to achieve a desirable property (like conditional coverage), but to demonstrate a failure mode of a widely-used metric (average interval length). Our primary contribution is to highlight this pitfall and propose a remedy (Interval Stability), a direction not explored by Barber et al. (2020).

## C  OMITTED PRELIMINARY

**Interval Prediction.** Interval prediction aims to construct a confidence interval that contains the true response value with a user-specified probability. Compared to traditional point estimation, interval prediction provides more comprehensive statistical information by quantifying the uncertainty using the interval length, which is often a more challenging goal. Definition 2 presents the formal definition.

**Definition 2** (Interval Prediction). *Let $(\boldsymbol{X}, Y)$ denote a feature-response pair. Given a miscoverage rate $\alpha$, interval prediction aims to construct a confidence interval $\mathcal{C}_{1-\alpha}(\boldsymbol{X})$, such that*

$$\mathbb{P}(Y \in \mathcal{C}_{1-\alpha}(\boldsymbol{X})) \geq 1 - \alpha. \tag{16}$$

*Given the coverage in Equation* (16)*, a smaller confidence interval indicates a more precise estimate.*

**Conformal Prediction.** To construct an interval prediction, we introduce a widely used approach called vanilla conformal prediction. The VCP method is typically divided into four stages: dataset splitting, training, calibration, and construction. The whole procedure is presented in Algorithm 2.

*Dataset Splitting.* Let $\mathcal{D} = \{(\boldsymbol{x}_i, y_i) : i \in \mathcal{I}\}$ denote the i.i.d. samples from a distribution $\mathcal{P}_{\boldsymbol{X}Y}$ over the covariate $\boldsymbol{X} \in \mathbb{R}^d$ and the response $Y \in \mathbb{R}$. The VCP first randomly splits the dataset $\mathcal{D}$ into

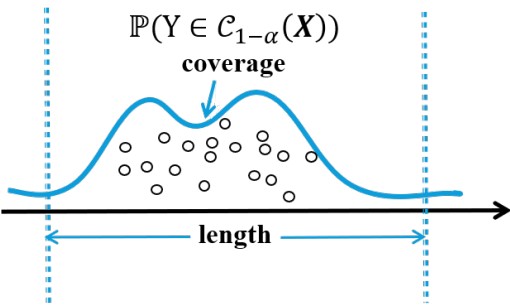

Figure 2: Illustration of coverage and interval length.

two folds: a training fold $\mathcal{D}_{\text{tr}} = \{(\boldsymbol{x}_i, y_i) : i \in \mathcal{I}_{\text{tr}}\}$ and a calibration fold $\mathcal{D}_{\text{ca}} = \{(\boldsymbol{x}_i, y_i) : i \in \mathcal{I}_{\text{ca}}\}$, where $\mathcal{I}_{\text{tr}} \cup \mathcal{I}_{\text{ca}} = \mathcal{I}$ and $\mathcal{I}_{\text{tr}} \cap \mathcal{I}_{ca} = \varnothing$.

*Training Process.* We train a model denoted by $\hat{\mu}(\cdot)$ (*e.g.*, a neural network) via the training fold $\mathcal{D}_{\text{tr}}$.

*Calibration Process.* Given the trained model $\hat{\mu}(\cdot)$, VCP calculates the non-conformity score on the calibration fold $\mathcal{D}_{\text{ca}}$, denoted by $\mathcal{V} = \{s(\boldsymbol{x}_i, y_i; \hat{\mu}) : i \in \mathcal{I}_{\text{ca}}\}$. The non-conformity score $s(\cdot)$ measures how well the model $\hat{\mu}(\cdot)$ fits the ground truth. A commonly used non-conformity score in regression tasks is the absolute residual, defined as $s(\boldsymbol{x}_i, y_i; \hat{\mu}) = |y_i - \hat{\mu}(\boldsymbol{x}_i)|$.

*Construction Process.* Finally, for a given miscoverage rate $\alpha$, we then compute a $(1-\tilde{\alpha})$-th quantile $\hat{Q}_{1-\tilde{\alpha}}(\mathcal{V})$ of the empirical distribution of the non-conformity score set $\mathcal{V}$ calculated on the calibration set, where $1 - \tilde{\alpha} = (1 - \alpha)(1 + 1/|\mathcal{V}|)$. The prediction interval at a new point $\boldsymbol{x}'$ is then given by

$$\mathcal{C}_{1-\alpha}(\boldsymbol{x}') = \{y : s(\boldsymbol{x}', y; \hat{\mu}) \leq \hat{Q}_{1-\tilde{\alpha}}(\mathcal{V})\}. \tag{17}$$

**Coverage and Length.** To evaluate the performance of interval prediction, two commonly used metrics: *coverage* and *length* are defined in Definition 3, as further illustrated in Figure 2.

**Definition 3** (Coverage and Length). *Let $(\boldsymbol{X}, Y)$ denote a feature-response pair from a joint distribution $\mathcal{P}_{\boldsymbol{X}Y}$, and let $\mathcal{C}_{1-\alpha}(\boldsymbol{X})$ denote the confidence interval to be evaluated and let $|\cdot|$ denote a certain measure of $\mathcal{C}_{1-\alpha}(\boldsymbol{X})$. The coverage and length of $\mathcal{C}_{1-\alpha}(\boldsymbol{X})$ is given by:*

$$\begin{aligned} Coverage &:= \mathbb{E}\left[\mathbb{I}(Y \in \mathcal{C}_{1-\alpha}(\boldsymbol{X}))\right], \\ Length &:= \mathbb{E}\left|\mathcal{C}_{1-\alpha}(\boldsymbol{X})\right|. \end{aligned} \tag{18}$$

*For example, the length of the prediction interval given by VCP in Equation* (17) *is:*

$$Length = \mathbb{E}\left[2\hat{Q}_{1-\tilde{\alpha}}(\mathcal{V})\right]. \tag{19}$$

Notably, the two metrics in Definition 3 evaluate the quality of prediction intervals from different perspectives. Figure 2 illustrates the coverage and length given a distribution. Firstly, high coverage ensures that the true value falls within the interval with high probability. A valid confidence interval should guarantee that the coverage exceeds $1 - \alpha$, as suggested in Equation 16. However, setting a sufficiently large interval always guarantees Equation (16), which is impractical and meaningless. Therefore, the length metric is required to ensure the interval's precision. Based on the above discussion, the gold standard in conformal prediction is *making the length as small as possible, given that the coverage is larger than $1 - \alpha$.*

Following the gold standard, VCP ensures the coverage guarantee under mild exchangeability assumption (Proposition 14), but pays less attention to the length. As a result, numerous works on improving the length of VCP from different perspectives (Papadopoulos et al., 2011; Romano et al., 2019) use intuitively valid approaches.

**Proposition 14** (Coverage Guarantee). *The terms $\mathcal{U}_i$ are exchangeable if arbitrary permutation leads to the same distribution, i.e., $(\mathcal{U}_1, ..., \mathcal{U}_{|\mathcal{I}_{ca}|+1}) \overset{d}{=} (\mathcal{U}_{\pi(1)}, ..., \mathcal{U}_{\pi(|\mathcal{I}_{ca}|+1)})$ with arbitrary permutation $\pi$ over $1, ..., |\mathcal{I}_{ca} + 1|$, where $\overset{d}{=}$ denotes equivalence in distribution. Suppose that the data pair $(\boldsymbol{x}_i, y_i), i \in \mathcal{I}_{ca}$ and the test point $(\boldsymbol{x}', y')$ are exchangeable, then the confidence interval $\mathcal{C}_{1-\alpha}(\boldsymbol{x}')$ returned by Algorithm 2 satisfies*

$$\mathbb{P}\left(y' \in \mathcal{C}_{1-\alpha}(\boldsymbol{x}')\right) \geq 1 - \alpha.$$

# D    PROOFS FOR THEOREMS AND COROLLARIES

## D.1    PROOF OF THEOREM 4

The returned interval could be in two folds: the vacuous fold and the meaningful fold. Therefore, by the definition of PT (Algorithm 1), we have

$$\mathbb{P}(y' \in \mathcal{C}_{1-\alpha}^{PT}(\boldsymbol{X}')) = \mathbb{P}(y' \in \mathcal{C}_{1-\alpha'}^{CP}(\boldsymbol{X}') \mid \mathcal{C}_{1-\alpha'}^{CP}(\boldsymbol{X}') \text{ is in the meaningful fold})p \tag{20}$$

$$+ \mathbb{P}(y' \in \mathcal{C}_{1-\alpha'}^{CP}(\boldsymbol{X}') \mid \mathcal{C}_{1-\alpha'}^{CP}(\boldsymbol{X}') \text{ is in the vacuous fold})(1-p) \tag{21}$$

$$\geq (1-\alpha')p = 1 - \alpha. \tag{22}$$

## D.2    PROOF OF THEOREM 5

Given that for all possible values of $\boldsymbol{X}$, there holds

$$\mathbb{P}(y \in \mathcal{C}_{1-\alpha}^{CP}(\boldsymbol{X}) \mid \boldsymbol{X}) \geq 1 - \alpha \tag{23}$$

holds almost surely, we have

$$\mathbb{P}(y' \in \mathcal{C}_{1-\alpha}^{PT}(\boldsymbol{X}') \mid \boldsymbol{X}) = \mathbb{P}(y' \in \mathcal{C}_{1-\alpha'}^{CP}(\boldsymbol{X}') \mid \mathcal{C}_{1-\alpha'}^{CP}(\boldsymbol{X}') \text{ is in the meaningful fold}, \boldsymbol{X})p \tag{24}$$

$$+ \mathbb{P}(y' \in \mathcal{C}_{1-\alpha'}^{CP}(\boldsymbol{X}') \mid \mathcal{C}_{1-\alpha'}^{CP}(\boldsymbol{X}') \text{ is in the vacuous fold}, \boldsymbol{X})(1-p) \tag{25}$$

$$\geq (1-\alpha')p = 1 - \alpha \tag{26}$$

holds almost surely.

## D.3    PROOF OF THEOREM 6

Given that it holds

$$\mathbb{P}(y \in \mathcal{C}_{1-\alpha}^{CP}(\boldsymbol{X}) \mid \boldsymbol{X} \in \mathcal{A}) = 1 - f_{\mathcal{A}}(\alpha), \tag{27}$$

we have

$$\mathbb{P}(y \in \mathcal{C}_{1-\alpha}^{PT}(\boldsymbol{X}) \mid \boldsymbol{X} \in \mathcal{A}) = p(1 - f_{\mathcal{A}}(\alpha')), \tag{28}$$

where $\alpha' = 1 - (1 - \alpha)/p$. And we have

$$\mathcal{F}(1) - \mathcal{F}(p) = f_{\mathcal{A}}(\alpha) - p f_{\mathcal{A}}\left(1 - \frac{1-\alpha}{p}\right) \geq 1 - p \tag{29}$$

$$\Rightarrow p\left(1 - f_{\mathcal{A}}\left(1 - \frac{1-\alpha}{p}\right)\right) \geq 1 - f_{\mathcal{A}}(\alpha) \tag{30}$$

$$\Rightarrow p(1 - \mathcal{A}(\alpha')) \geq 1 - f_{\mathcal{A}}(\alpha). \tag{31}$$

Therefore, we have

$$\mathbb{P}(y \in \mathcal{C}_{1-\alpha}^{PT}(\boldsymbol{X}) \mid \boldsymbol{X} \in \mathcal{A}) \geq \mathbb{P}(y \in \mathcal{C}_{1-\alpha}^{CP}(\boldsymbol{X}) \mid \boldsymbol{X} \in \mathcal{A}). \tag{32}$$

## D.4    PROOF OF THEOREM 7

By the definition of $\mathcal{L}$ and PT in Algorithm 1, when the returned set belongs to the meaningful fold, the length is $\mathcal{L}(\boldsymbol{x}, (1 - \tilde{\alpha})/p; s)$, given the non-conformity score function and miscoverage rate $\tilde{\alpha}$. And the length of the returned set belongs to the vacuous fold is $0$. Therefore, the expected length of the set returned by PT is

$$p\mathbb{E}\left(\mathcal{L}\left(\boldsymbol{x}, \frac{1-\tilde{\alpha}}{p}; s\right)\right), \tag{33}$$

where the expectation is taken over $\boldsymbol{x}$. Then we get the general sufficient condition is

$$\exists p \in (1 - \tilde{\alpha}, 1) \quad s.t. \quad p\mathbb{E}\left(\mathcal{L}\left(\boldsymbol{x}, \frac{1-\tilde{\alpha}}{p}; s\right)\right) < \mathbb{E}(\mathcal{L}(\boldsymbol{x}, 1 - \tilde{\alpha}; s)). \tag{34}$$

## D.5 PROOF OF THEOREM 8

Let $\mathcal{G}(c) = \mathbb{E}(\mathcal{L}(\boldsymbol{x}, 1 - \tilde{\alpha}; s))$, the sufficient condition in Theorem 7 is

$$\exists p \in (c, 1), \ s.t. \ p\mathcal{G}(c/p) < \mathcal{G}(c). \tag{35}$$

Note that when $p = 1$, $p\mathcal{G}(c/p) = \mathcal{G}(c)$, the sufficient condition of Eq (35) is

$$\mathcal{F}'(1) > 0, \ \text{where} \ \mathcal{F}(p) = p\mathcal{G}(c/p), \tag{36}$$

which is equivalent to

$$\frac{\mathcal{G}(c)}{c} > \mathcal{G}'(c) \Rightarrow \mathbb{E}\left(\frac{\mathcal{L}(\boldsymbol{x}, 1 - \tilde{\alpha}; s)}{1 - \tilde{\alpha}}\right) > \mathbb{E}\left(\left.\frac{\partial}{\partial \alpha}\mathcal{L}(\boldsymbol{x}, \alpha; s)\right|_{\alpha = 1 - \tilde{\alpha}}\right). \tag{37}$$

## D.6 PROOF OF THEOREM 11

Use the notation in Appendix D.5, the general sufficient condition could be written as

$$\exists p \in (c, 1), \ s.t. \ p\mathcal{G}(c/p) < \mathcal{G}(c). \tag{38}$$

If there exists $u \in (1 - \tilde{\alpha})$ satisfies

$$\frac{\mathcal{G}(c)}{c} > \frac{\mathcal{G}(u) - \mathcal{G}(c)}{u - c}, \tag{39}$$

we have

$$\frac{c}{u}\mathcal{G}(u) < \mathcal{G}(c). \tag{40}$$

Let $p = c/u$, we have

$$p\mathcal{G}(c/p) < \mathcal{G}(c). \tag{41}$$

Eq (39) is actually the condition

$$\frac{\mathbb{E}(\mathcal{L}(\boldsymbol{x}, 1 - \tilde{\alpha}; s))}{1 - \tilde{\alpha}} > \frac{\mathbb{E}(\mathcal{L}(\boldsymbol{x}, u; s)) - \mathbb{E}(\mathcal{L}(\boldsymbol{x}, 1 - \tilde{\alpha}; s))}{u - (1 - \tilde{\alpha})}. \tag{42}$$

## D.7 PROOF OF EXAMPLE 12

When the non-conformity score follows a Gaussian distribution, the analytical solutions of the interval length returned by VCP and PT-VCP are

$$|\mathcal{C}_{1-\tilde{\alpha}}^{VCP}(\boldsymbol{X}')| = 2\Phi^{-1}\left(1 - \frac{\tilde{\alpha}}{2}\right), \quad |\mathcal{C}_{1-\tilde{\alpha}}^{PT}(\boldsymbol{X}')| = 2p\Phi^{-1}\left(1 - \frac{1}{2}\left(1 - \frac{1 - \tilde{\alpha}}{p}\right)\right) \tag{43}$$

where $\Phi(\cdot)$ is the cumulative distribution function of the Gaussian distribution. Therefore, PT fails since Lemma 1.

**Lemma 1.** $\forall \alpha \in (0, 1), p \in (1 - \alpha, 1)$, *there holds*

$$\Phi^{-1}\left(1 - \frac{\tilde{\alpha}}{2}\right) < p\Phi^{-1}\left(1 - \frac{1}{2}\left(1 - \frac{1 - \tilde{\alpha}}{p}\right)\right) \tag{44}$$

*Proof.* Using the symmetry identity $\Phi^{-1}(1 - u) = -\Phi^{-1}(u)$, the desired inequality is equivalent to

$$p\,\Phi^{-1}\left(\tfrac{1}{2}\left(1 - \tfrac{1-\alpha}{p}\right)\right) < \Phi^{-1}(\alpha/2). \tag{45}$$

Define

$$u_0 := \alpha/2 \in (0, 1/2), \qquad u(p) := \tfrac{1}{2}\left(1 - \tfrac{1-\alpha}{p}\right) \in (0, 1/2). \tag{46}$$

Let $g(u) := \Phi^{-1}(u)$ on $(0, 1/2)$. Since

$$g'(u) = \frac{1}{\phi(\Phi^{-1}(u))} > 0, \qquad g''(u) = \frac{\Phi^{-1}(u)}{\phi(\Phi^{-1}(u))^2} < 0, \tag{47}$$

where $\phi(\cdot)$ denotes the p.d.f. of the standard normal distribution, the function $g$ is increasing and strictly concave. Hence, for any $u \leq u_0$,

$$g(u) \ \leq \ g(u_0) + g'(u_0)\,(u - u_0). \tag{48}$$

Plugging $u = u(p)$ into Eq (48) and multiplying both sides by $p \in (0, 1)$ gives

$$p\,g(u(p)) \ \leq \ p\,g(u_0) \ + \ p\,g'(u_0)\,\big(u(p) - u_0\big). \tag{49}$$

A direct calculation yields

$$u(p) - u_0 = \frac{(1-\alpha)(p-1)}{2p} \ < \ 0. \tag{50}$$

Subtracting $g(u_0)$ from both sides of Eq (49) and using Eq (50), we obtain

$$p\,g(u(p)) - g(u_0) \ \leq \ (1-p)\Big[-g(u_0) + \frac{1-\alpha}{2\,\phi(g(u_0))}\Big]. \tag{51}$$

Here $1 - p > 0$. Moreover, since $g(u_0) = \Phi^{-1}(\alpha/2) < 0$ and $\phi(g(u_0)) > 0$, the bracket in Eq (51) is nonnegative. Thus the right-hand side of Eq (51) is nonpositive, implying

$$p\,g(u(p)) - g(u_0) < 0, \tag{52}$$

which is exactly inequality Eq (45). The inequality is strict because $u(p) \neq u_0$ and $g$ is strictly concave.

Reverting to the upper-tail form via $\Phi^{-1}(1 - u) = -\Phi^{-1}(u)$ completes the proof:

$$\Phi^{-1}(1 - \alpha/2) \ < \ p\,\Phi^{-1}\Big(\tfrac{1}{2}\Big(1 + \tfrac{1-\alpha}{p}\Big)\Big). \tag{53}$$

$\square$

## E  MISSING ILLUSTRATION

In this section, we present the missing illustration of Example 1 in Section 1, the illustration of PT (Figure 4) and VCP algorithm mentioned in Section 3.1.

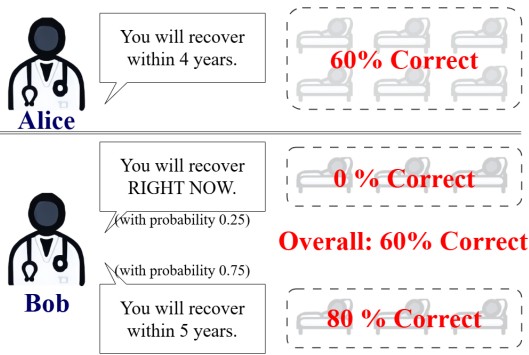

Figure 3: Illustration of Example 1. Doctor Alice and Bob both achieve $60\%$ accuracy. Bob is more precise regarding length, but the corresponding strategy is not practically valid.

## F  OMITTED EXPERIMENTS

In this section, we present all the omitted experiments. In Appendix F.1, we demonstrate the missing experimental results in Section 3.3 and Section 3.4. In Appendix F.2, we exhibit the ablation study results.

---

**Algorithm 2** Vanilla Conformal Prediction (VCP)

---

1: **Input:** miscoverage rate $\alpha$, dataset $\mathcal{D} = \{(\boldsymbol{x}_i, y_i) : i \in \mathcal{I}\}$, test point $\boldsymbol{x}'$, non-conformity score function $s(\boldsymbol{x}_i, y_i; \hat{\mu})$.
2: Randomly split $\mathcal{D}$ into a training fold $\mathcal{D}_{\text{tr}} = \{(\boldsymbol{x}_i, y_i) : i \in \mathcal{I}_{\text{tr}}\}$ and a calibration fold $\mathcal{D}_{\text{ca}} = \{(\boldsymbol{x}_i, y_i) : i \in \mathcal{I}_{\text{ca}}\}$;
3: Train a model $\hat{\mu}$ based on the training fold $\mathcal{D}_{\text{tr}}$;
4: Calculate the non-conformity score on the calibration fold $\mathcal{D}_{\text{ca}}$, denoted by $\mathcal{V} = \{s(\boldsymbol{x}_i, y_i, \hat{\mu}) : i \in \mathcal{I}_{\text{ca}}\}$;
5: Compute the $(1 - \tilde{\alpha})$-th quantile $\hat{Q}_{1-\tilde{\alpha}}(\mathcal{V})$ of the empirical distribution of the non-conformity score set $\mathcal{V}$ calculated on the calibration set $\mathcal{D}_{\text{ca}}$, where $1 - \tilde{\alpha} = (1 - \alpha)(1 + 1/|\mathcal{V}|)$;
6: **Output:** Interval $\mathcal{C}_{1-\alpha}(\boldsymbol{x}') = \{y : s(\boldsymbol{x}', y; \hat{\mu}) \leq \hat{Q}_{1-\tilde{\alpha}}(\mathcal{V})\}$.

---

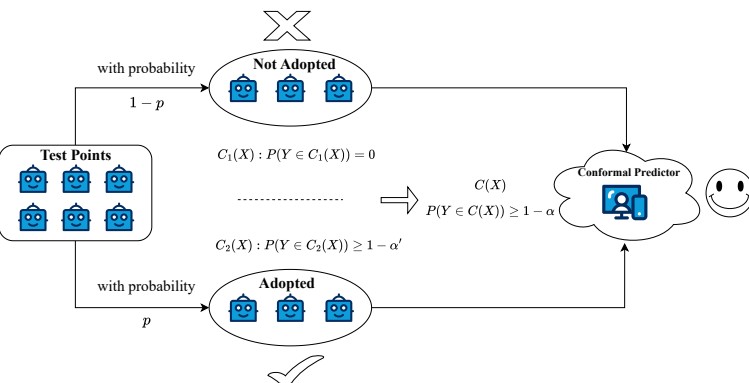

Figure 4: The illustration of Prejudicial Trick (PT). To obtain a $1 - \alpha$ confidence interval, PT first assigns empty sets for a $1 - p$ subset of the test points, and assigns $1 - \alpha'$ confidence interval for the remaining test points where $\alpha' < \alpha$. The returned confidence interval still satisfies $\mathbb{P}(Y \in \mathcal{C}(X)) \geq 1 - \alpha$ by setting a proper $\alpha'$.

### F.1 OMITTED EXPERIMENTAL RESULTS

**Classification Tasks.** We extend PT to classification tasks. We apply PT to the real-world IMAGENET-VAL dataset (Deng et al., 2009) with several pre-trained models, a similar setting with Angelopoulos et al. (2020). To simulate model misspecification, a bias is introduced to the logits of several classes before the softmax operation. The magnitude of the bias is determined based on the scale of the outputs. The experimental results on classification tasks in Table 4 perform similarly to regression tasks. Specifically, PT-VCP attains valid coverage across different models (Theorem 4) while improving the length compared to VCP (Theorem 11).

**Conformalized Quantile Regression.** We deploy PT into other variants of conformal prediction. Specifically, we choose CQR as a baseline (Romano et al., 2019). CQR inherits the advantages of both conformal prediction and classical quantile regression. We use the same datasets and evaluation metrics as in Section 3.4. To mimic the model misspecification, we add bias directly to the lower and upper quantiles obtained by the quantile regression. The experimental results on the real-world CQR tasks are exhibited in Table 5. It illustrates that *PT achieves shorter interval length while maintaining valid coverage on CQR*.

**Group Coverage.** In Section 3.3, we conduct experiments to evaluate the different performance of VCP and PT-VCP regarding group coverage. The experimental results shown in Table 6 demonstrate that PT not only achieves shorter confidence intervals while maintaining overall coverage, *but also improves the group coverage in regression tasks*[4].

---

[4]Group coverage is defined as the lowest coverage rate among all the groups.

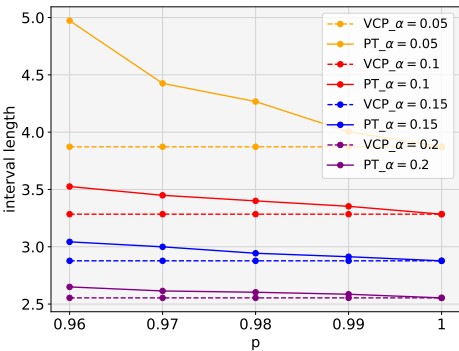

Figure 5: PT fails to improve length when the conditions on the distribution of non-conformity score are not safisfied.

Table 4: Comparison between VCP and PT-VCP in classification tasks across different models. Experiments on RAPS ($\alpha = 0.1, p = 0.95$), with index range chosen as $300$.

| METHOD | BIAS | VCP | | PT-VCP | |
|---|---|---|---|---|---|
| MODEL | | COVERAGE | LENGTH | COVERAGE | LENGTH |
| RESNET18 | 40 | 0.90 ±0.000 | 304.02 ±0.004 | 0.90 ±0.000 | **295.60** ±0.233 |
| RESNET50 | 40 | 0.90 ±0.000 | 302.09 ±0.027 | 0.90 ±0.000 | **290.29** ±0.224 |
| RESNET101 | 40 | 0.90 ±0.000 | 302.01 ±0.004 | 0.90 ±0.000 | **289.56** ±0.174 |
| RESNET152 | 40 | 0.89 ±0.004 | 301.53 ±0.054 | 0.90 ±0.000 | **288.98** ±0.165 |
| RESNEXT101 | 40 | 0.90 ±0.000 | 301.48 ±0.013 | 0.90 ±0.000 | **288.49** ±0.201 |
| VGG16 | 40 | 0.90 ±0.004 | 303.39 ±0.143 | 0.90 ±0.000 | **293.34** ±0.304 |
| SHUFFLENET | 40 | 0.90 ±0.000 | 304.05 ±0.040 | 0.90 ±0.000 | **295.79** ±0.282 |
| INCEPTION | 40 | 0.90 ±0.000 | 304.10 ±0.013 | 0.90 ±0.000 | **297.25** ±0.228 |
| DENSENET161 | 40 | 0.90 ±0.000 | 302.03 ±0.009 | 0.90 ±0.000 | **289.29** ±0.197 |

**Interval Stability.** In Section 4, we introduce a new evaluation criterion, termed *interval stability* and conduct several empirical evaluations using the datasets described in Section 3.4, the results of which are listed in Table 3. We further investigate the performance of the interval stability metric using CQR as the base algorithm in Table 7. The results are similar to the results in Table 3. We also evaluate the interval stability metric on classification tasks. As shown in Table 8, interval stability successfully identify the vacuous randomness in PT.

### F.2 ABLATION STUDIES

This section exhibits the ablation studies on the probability hyperparameter $p$ in PT and the bias parameter $\mu$ on different base algorithms (Figure 6-Figure 15). All the experiments are conducted based on various miscoverage rates $\alpha$. The experiment results demonstrate that, although not all the probability hyperparameters $p$ outperform the base algorithm, our goal is to show that *there exist multiple (at least one) probability hyperparameters such that PT-VCP outperforms VCP, which suffices to challenge the coverage-length gold standard.* Furthermore, we find that the bias parameter actually matters here, implying that PT-VCP performs better than VCP under misspecification, which validates Theorem 11.

## G EXPERIMENT DETAILS

In this section, we provide implementation details of the experiments in this paper, including experiments on synthetic datasets in Appendix G.1 and experiments on real-world datasets in Appendix G.2.

Table 5: Comparison between CQR and PT-CQR in quantile regression task across different datasets.

| Method | Bias | CQR | | PT-CQR | |
|---|---|---|---|---|---|
| Dataset | | Coverage | Length | Coverage | Length |
| meps-19 | 1 | 0.91 ±0.000 | 4.60 ±0.148 | 0.91 ±0.246 | **4.44** ±0.143 |
| meps-20 | 1 | 0.91 ±0.000 | 4.58 ±0.192 | 0.91 ±0.179 | **4.41** ±0.188 |
| meps-21 | 1 | 0.91 ±0.000 | 4.65 ±0.080 | 0.91 ±0.161 | **4.52** ±0.107 |
| bike | 1 | 0.91 ±0.000 | 2.61 ±0.013 | 0.90 ±0.268 | **2.51** ±0.009 |
| blog-data | 1 | 0.91 ±0.000 | 3.80 ±0.107 | 0.93 ±0.116 | **3.61** ±0.098 |
| bio | 1 | 0.91 ±0.000 | 3.45 ±0.009 | 0.90 ±0.112 | **3.32** ±0.009 |
| facebook-1 | 1 | 0.91 ±0.000 | 3.38 ±0.022 | 0.92 ±0.125 | **3.22** ±0.027 |
| facebook-2 | 1 | 0.91 ±0.000 | 3.57 ±0.027 | 0.92 ±0.085 | **3.39** ±0.027 |
| concrete | 2 | 0.91 ±0.000 | 4.39 ±0.022 | 0.88 ±0.648 | **4.23** ±0.018 |
| star | 2 | 0.91 ±0.000 | 4.15 ±0.004 | 0.90 ±0.349 | **3.96** ±0.009 |

Table 6: Comparison of group coverage between VCP and PT-VCP on regression tasks across different datasets ($\alpha = 0.1$).

| Dataset | Group | VCP | PT-VCP |
|---|---|---|---|
| bike | Day | 0.878 ± 0.007 | **0.884** ± 0.010 |
| | Month | 0.826 ± 0.010 | **0.857** ± 0.011 |
| | Year | 0.851 ± 0.005 | **0.871** ± 0.004 |
| star | Gender | 0.905 ± 0.008 | **0.905** ± 0.002 |
| | Stark | 0.890 ± 0.005 | **0.895** ± 0.007 |
| | School1 | 0.902 ± 0.008 | **0.899** ± 0.022 |
| meps-19 | SEX=1 | 0.883 ± 0.004 | **0.895** ± 0.001 |
| | MARRY=1 | 0.901 ± 0.004 | **0.901** ± 0.003 |
| | REGION=1 | 0.862 ± 0.005 | **0.877** ± 0.006 |
| meps-20 | FTSTU=1 | 0.893 ± 0.004 | **0.900** ± 0.002 |
| | ACTDTY=1 | 0.897 ± 0.003 | **0.902** ± 0.002 |
| | HONRDC=1 | 0.792 ± 0.010 | **0.846** ± 0.008 |
| meps-21 | RTHLTH=1 | 0.864 ± 0.004 | **0.877** ± 0.004 |
| | MNHLTH=1 | 0.856 ± 0.004 | **0.873** ± 0.004 |
| | HIBPDX=1 | 0.755 ± 0.013 | **0.818** ± 0.009 |

## G.1 SYNTHETIC DATASETS

This section presents experiment details about the motivating example (Section 3.2) in Appendix G.1.1 and failure case (Section 3.4) in Appendix G.1.2.

### G.1.1 MOTIVATING EXAMPLE

In our motivating example, we consider a simple data-generating process where the true underlying model is linear with Gaussian mixture noise:

$$Y = \boldsymbol{X}^\top \boldsymbol{\beta} + \epsilon, \quad \boldsymbol{X} \sim \mathcal{N}(\boldsymbol{0}, I_2).$$

The noise term $\epsilon$ follows $\mathcal{N}(\mu, 1)$ with probability 0.5 and $\mathcal{N}(-\mu, 1)$ with probability 0.5. The training, calibration, and test folds are all generated from this distribution. To emulate model misspecification, we fit the training fold using a linear model with Gaussian noise. Throughout the experiments, we set $\mu = 20$, $\alpha \in \{0.1, 0.2\}$, and $p \in \{0.96, 0.98\}$. We further average results over 5 random seeds and report the corresponding standard errors. Both VCP (Algorithm 2) and PT-VCP are evaluated under this setting.

Table 7: Comparison between CQR and PT-CQR regarding interval stability.

| Dataset | CQR | PT-CQR |
|---|---|---|
| meps-19 | **0.00** $\pm$0.000 | 0.13 $\pm$0.005 |
| meps-20 | **0.00** $\pm$0.000 | 0.13 $\pm$0.006 |
| meps-21 | **0.00** $\pm$0.000 | 0.14 $\pm$0.003 |
| bike | **0.00** $\pm$0.000 | 0.08 $\pm$0.001 |
| blog-data | **0.00** $\pm$0.000 | 0.11 $\pm$0.003 |
| bio | **0.00** $\pm$0.000 | 0.10 $\pm$0.000 |
| facebook-1 | **0.00** $\pm$0.000 | 0.10 $\pm$0.001 |
| facebook-2 | **0.00** $\pm$0.000 | 0.10 $\pm$0.001 |
| concrete | **0.00** $\pm$0.000 | 0.13 $\pm$0.002 |
| star | **0.00** $\pm$0.000 | 0.12 $\pm$0.001 |

Table 8: Comparison between VCP and PT-VCP in classification tasks regarding interval stability. Experiments on RAPS ($\alpha = 0.1, p = 0.95$), with index range chosen as $300$.

| MODEL | BIAS | VCP | PT-VCP |
|---|---|---|---|
| RESNET18 | 40 | **0.02** $\pm$0.004 | 12.08 $\pm$0.060 |
| RESNET50 | 40 | **0.07** $\pm$0.017 | 11.86 $\pm$0.057 |
| RESNET101 | 40 | **0.01** $\pm$0.004 | 11.93 $\pm$0.089 |
| RESNET152 | 40 | **0.19** $\pm$0.002 | 11.91 $\pm$0.058 |
| RESNEXT101 | 40 | **0.20** $\pm$0.000 | 11.89 $\pm$0.083 |
| VGG16 | 40 | **0.11** $\pm$0.030 | 12.00 $\pm$0.060 |
| SHUFFLENET | 40 | **0.03** $\pm$0.025 | 12.08 $\pm$0.055 |
| INCEPTION | 40 | **0.07** $\pm$0.010 | 12.19 $\pm$0.080 |
| DENSENET161 | 40 | **0.03** $\pm$0.006 | 11.90 $\pm$0.057 |

### G.1.2 FAILURE CASE

In Figure 5, we present a failure case where VCP outperforms PT-VCP. Here, we modify the data-generating process to a linear model with Gaussian noise and fit it with the same linear model, so that no model misspecification arises in contrast to our motivating example. In this setting, the distribution of the score function fails to satisfy the sufficient condition in Theorem 11, which explains why VCP outperforms PT-VCP.

### G.2 REAL WORLD DATASETS

In this section, we firstly introduce the model structure in our experiments in Appendix G.2.1. Then we present the experiment details, including experiments on marginal and group coverage (Appendix G.2.2, Appendix G.2.6), regression tasks (Appendix G.2.3), classification tasks (Appendix G.2.4), ablation studies (Appendix G.2.5) and interval stability (Appendix G.2.7).

### G.2.1 MODEL STRUCTURE

In this section, we present the details of the structure of our model on real world datasets. Specifically, our model shares the same structure with(Romano et al., 2019).

**Neural Net.** Our neural network design includes three fully connected layers, with ReLU activation functions applied between each layer. The initial layer accepts an input feature vector $X$ of n dimensions and produces $64$ hidden units. The second layer mirrors this structure, generating another set of $64$ hidden units. The final layer is a linear output layer that provides a pointwise prediction for the response variable $Y$. The network's parameters are optimized by minimizing a quadratic loss function. We used the Adam optimization algorithm with a constant learning rate of $5 \times 10^{-4}$, minibatch size of $64$, and a weight decay coefficient of $10^{-6}$. In addition, regularization of dropout is implemented, with a retention probability of $0.1$ for hidden units. To avoid overfitting, early stop is used and the number of training epochs is determined by cross-validation, with a maximum cap of 1000 epochs.

**CQR Neural Net.** We utilize neural networks to implement CQR for quantile regression. The net-

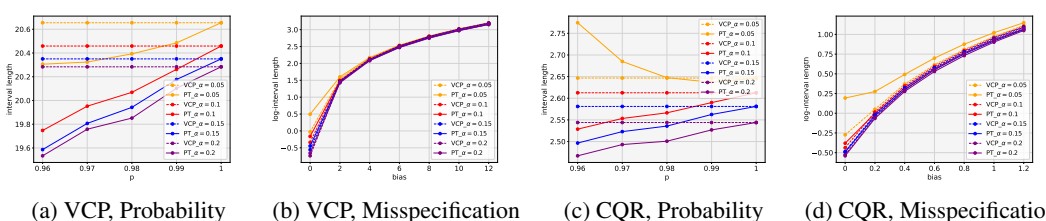

| (a) VCP, Probability | (b) VCP, Misspecification | (c) CQR, Probability | (d) CQR, Misspecification |

Figure 6: Ablation studies of dataset BIKE on different misspecification levels (b, d) and probability hyperparameters (a, c), including comparisons with VCP (a–b) and CQR (c–d).

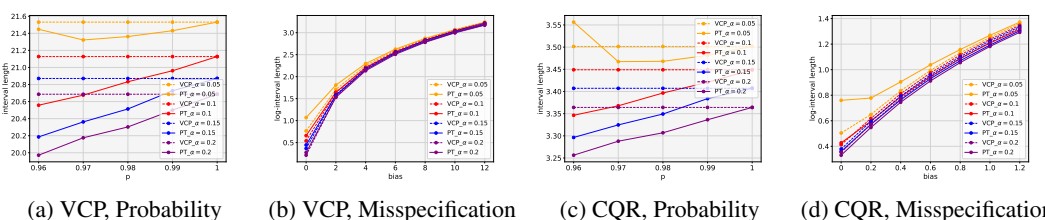

| (a) VCP, Probability | (b) VCP, Misspecification | (c) CQR, Probability | (d) CQR, Misspecification |

Figure 7: Ablation studies of dataset BIO on different misspecification level (a, c) and probability hyperparameter (b, d), including the comparison with VCP (a-b) and CQR (c-d).

work structure is consistent with the one described above, with the sole difference being that the output of the quantile regression network is a two-dimensional vector, which indicates the lower and upper conditional quantiles. Additionally, the training process remains the same, except that the pinball loss function in equation is employed instead of the quadratic loss.

### G.2.2 MARGINAL AND GROUP COVERAGE

In Figure 1, we compare the coverage of VCP and PT-VCP on the BIKE dataset (Fanaee-T, 2013). Specifically, we evaluate several choices of $\alpha$ and $p$, and report both the marginal coverage and the group coverage (an empirical indicator of conditional coverage). The group coverage is obtained by partitioning the data according to Day, Month, and Year.

### G.2.3 REGRESSION TASKS

In ordinary regression tasks, we employ the neural network described in Section G.2.1 to fit several real-world datasets: MEPS19–21 (Cohen et al., 2009), BIKE (Fanaee-T, 2013), BLOG-DATA (Buza, 2014), BIO (Rana, 2013), FACEBOOK1–2 (Singh, 2015), CONCRETE (Yeh, 1998), and STAR (Achilles et al., 2008). To mimic model misspecification, we introduce a bias term that is directly added to the logits output by the neural network. The magnitude of this bias term, which varies across datasets, is reported in Table 2. Throughout these experiments, we set $\alpha = 0.1$ and $p = 0.95$, under which both VCP (Algorithm 2) and PT-VCP are evaluated. We further average results over 5 random seeds and report the corresponding standard errors.

In CQR tasks, we employ the CQR neural network described in Section G.2.1 to fit several real-world datasets: MEPS19–21 (Cohen et al., 2009), BIKE (Fanaee-T, 2013), BLOG-DATA (Buza, 2014), BIO (Rana, 2013), FACEBOOK1–2 (Singh, 2015), CONCRETE (Yeh, 1998), and STAR (Achilles et al., 2008). To mimic model misspecification, we introduce a bias term by directly adding it to both the lower and upper quantiles estimated by the quantile regression. The magnitude of this bias varies across datasets. Throughout the experiments, we set $\alpha = 1$ and $p = 0.95$, under which both CQR and PT-CQR are evaluated, as reported in Table 5. We further average results over 5 random seeds and report the corresponding standard errors.

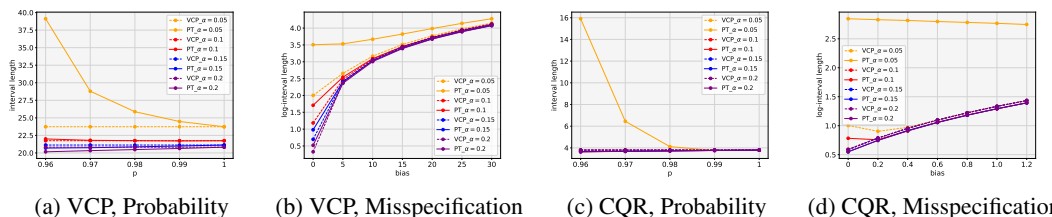

(a) VCP, Probability     (b) VCP, Misspecification     (c) CQR, Probability     (d) CQR, Misspecification

Figure 8: Ablation studies of dataset BLOGDATA on different misspecification level (a, c) and probability hyperparameter (b, d), including the comparison with VCP (a-b) and CQR (c-d).

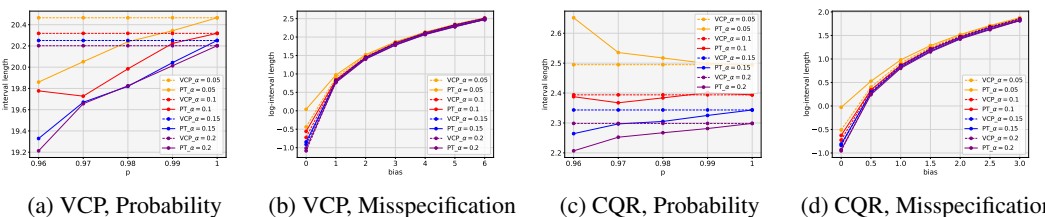

(a) VCP, Probability     (b) VCP, Misspecification     (c) CQR, Probability     (d) CQR, Misspecification

Figure 9: Ablation studies of dataset CONCRETE on different misspecification level (a, c) and probability hyperparameter (b, d), including the comparison with VCP (a-b) and CQR (c-d).

### G.2.4 CLASSIFICATION TASK

In classification tasks, we apply PT to the real-world IMAGENET-VAL dataset (Deng et al., 2009) using several pre-trained models listed in Table 4, following a setting similar to Angelopoulos et al. (2020). To simulate model misspecification, we introduce a bias to the logits of several classes before the softmax operation. The magnitude of this bias is scaled according to the outputs, and the number of biased classes is specified by an index range in our experiments. For the classification task, we adopt RAPS (Angelopoulos et al., 2020) as the score function and set $\alpha = 0.1$ and $p = 0.95$. We further average results over 5 random seeds and report the corresponding standard errors.

### G.2.5 ABLATION STUDIES

The ablation studies mainly focus on regression tasks, including both ordinary regression and CQR. We conduct experiments on all datasets used in the regression setting. For these ablation studies, we set $\alpha \in \{0.05, 0.1, 0.15, 0.2\}$, $p \in \{0.96, 0.97, 0.98, 0.99, 1.00\}$, and apply dataset-specific bias magnitudes.

### G.2.6 GROUP COVERAGE

To demonstrate that PT-VCP does not degrade conditional coverage compared to VCP, we conduct experiments measuring group coverage on the MEPS19–21 (Cohen et al., 2009), BIKE (Fanaee-T, 2013), and STAR (Achilles et al., 2008) datasets. The grouping strategies are summarized in Table 6. In these experiments, we set $\alpha = 0.1$ and $p = 0.95$, and further average results over 5 random seeds, reporting the corresponding standard errors.

### G.2.7 INTERVAL STABILITY

To compare the newly proposed metric *interval stability* between VCP and PT-VCP, we conduct experiments on both regression and classification tasks, using different datasets and pre-trained models, respectively. The results are reported in Table 3, Table 7, and Table 8. We measure *interval stability* by computing the variance of the interval (or set) length (or size) for repeated predictions on the same input, and then averaging this variance over all inputs $X$. Results are further averaged over 5 random seeds, with the corresponding standard errors reported.

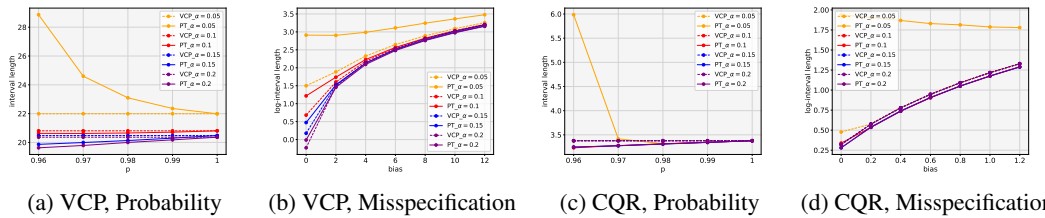

(a) VCP, Probability    (b) VCP, Misspecification    (c) CQR, Probability    (d) CQR, Misspecification

Figure 10: Ablation studies of dataset FACEBOOK1 on different misspecification level (a, c) and probability hyperparameter (b, d), including the comparison with VCP (a-b) and CQR (c-d).

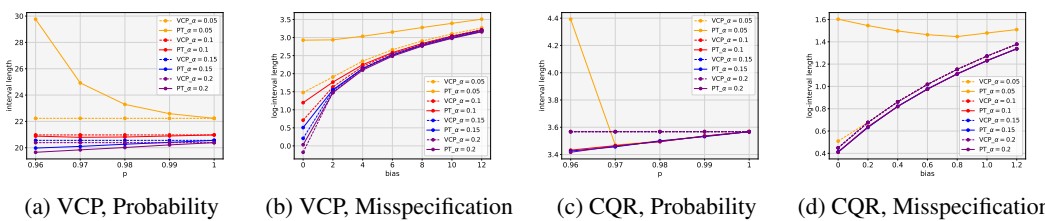

(a) VCP, Probability    (b) VCP, Misspecification    (c) CQR, Probability    (d) CQR, Misspecification

Figure 11: Ablation studies of dataset FACEBOOK2 on different misspecification level (a, c) and probability hyperparameter (b, d), including the comparison with VCP (a-b) and CQR (c-d).

## H  ADDITIONAL RELATED WORKS

**Conformal prediction.** Conformal prediction is a post hoc calibration framework that constructs statistically rigorous uncertainty sets for predictions from machine learning models (Vovk et al., 2005; Shafer & Vovk, 2008; Lei et al., 2018; Foygel Barber et al., 2021; Angelopoulos & Bates, 2021; Papadopoulos et al., 2008). Traditionally, vanilla conformal prediction is deployed in regression tasks (Vovk et al., 2005; Shafer & Vovk, 2008; Lei et al., 2018). Later, a branch of research expands vanilla conformal prediction to diverse data structures and applications, including classification tasks (Angelopoulos et al., 2020; Dabah & Tirer, 2024), censored data in survival analysis (Teng et al., 2021; Candès et al., 2023), functional data (Lei et al., 2015; Ajroldi et al., 2023), graph-based models (Zargarbashi et al., 2023; Zargarbashi & Bojchevski, 2024), time series data (Xu & Xie, 2021; Stankeviciute et al., 2021), treatment effects (Lei & Candès, 2021; Jin et al., 2023), *etc*.

**Interval regression.** While coverage guarantees and interval length serve as fundamental metrics for evaluating conformal prediction (Vovk et al., 2005; Lei et al., 2018; Barber et al., 2020), these criteria are deeply entrenched in the broader paradigm of interval regression methodologies. Established approaches including quantile regression (Alaa et al., 2023; Sasaki et al., 2022) and Bayesian credible intervals (Kuleshov et al., 2018; Wang & Ghosal, 2023) similarly prioritize the dual metrics. Of particular relevance is Navratil et al. (2020) who proposes an excess and deficit metrics beyond the traditional coverage-length metric. Our paper differs from Navratil et al. (2020) in that our main contributions center on uncovering the inherent limitations of coverage-length metrics. Additionally, we contend that the proposed excess and deficit metrics cannot be directly applied to PT-VCP.

## I  THE USE OF LARGE LANGUAGE MODELS (LLMS)

In preparing this paper, a large language model (LLM) was employed solely for language refinement purposes, such as improving the clarity and fluency of expressions. The LLM did not contribute to research ideation, methodology, data analysis, or substantive content generation. The authors fully acknowledge responsibility for all contents of the paper, including any text polished with the assistance of the LLM.

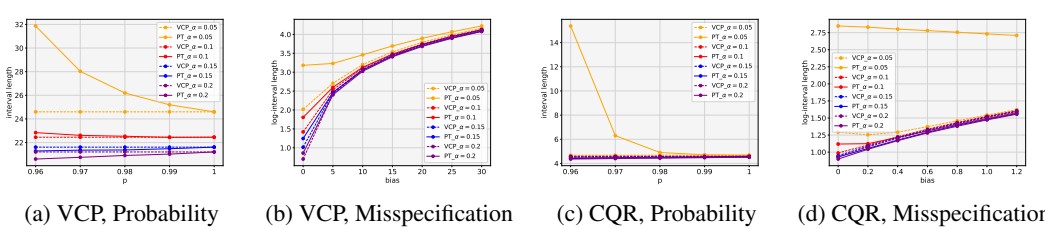

(a) VCP, Probability     (b) VCP, Misspecification     (c) CQR, Probability     (d) CQR, Misspecification

Figure 12: Ablation studies of dataset MEPS19 on different misspecification level (a, c) and probability hyperparameter (b, d), including the comparison with VCP (a-b) and CQR (c-d).

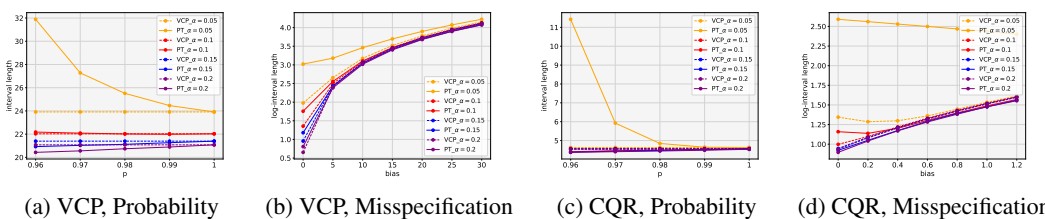

(a) VCP, Probability     (b) VCP, Misspecification     (c) CQR, Probability     (d) CQR, Misspecification

Figure 13: Ablation studies of dataset MEPS20 on different misspecification level (a, c) and probability hyperparameter (b, d), including the comparison with VCP (a-b) and CQR (c-d).

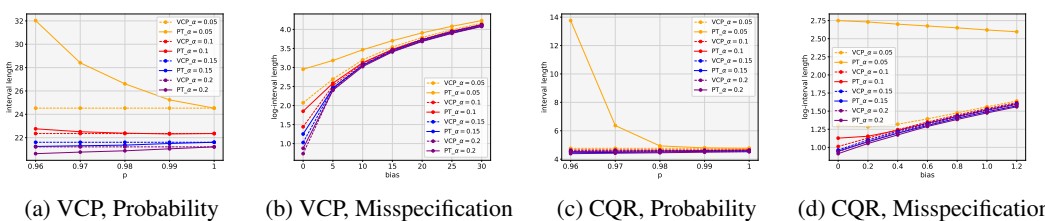

(a) VCP, Probability     (b) VCP, Misspecification     (c) CQR, Probability     (d) CQR, Misspecification

Figure 14: Ablation studies of dataset MEPS21 on different misspecification level (a, c) and probability hyperparameter (b, d), including the comparison with VCP (a-b) and CQR (c-d).

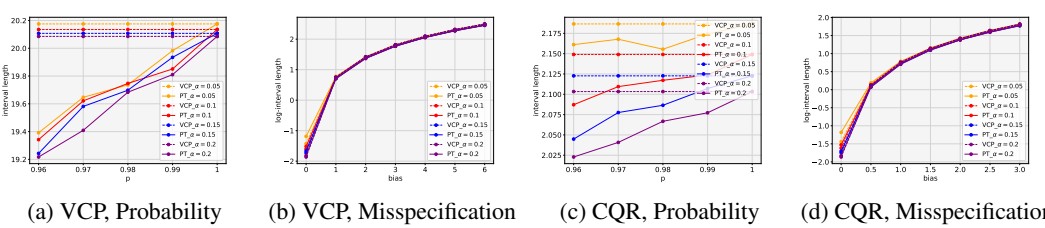

(a) VCP, Probability     (b) VCP, Misspecification     (c) CQR, Probability     (d) CQR, Misspecification

Figure 15: Ablation studies of dataset STAR on different misspecification level (a, c) and probability hyperparameter (b, d), including the comparison with VCP (a-b) and CQR (c-d).

