# OpenReview forum: "A Pitfall in Conformal Prediction:  When Shorter Intervals Are Not Better"
_ICLR.cc/2026/Conference — Submitted to ICLR 2026_

### Official Review · Reviewer_mhUR · 2025-10-28

**Soundness:** 2
**Presentation:** 2
**Contribution:** 2
**Rating:** 2
**Confidence:** 5

**Summary:**

The paper is clearly written and easy to follow.

Problem context: Conventionally, CP methods are evaluated by its coverage and interval length. This work argues the sufficiency of these standard metrics.

Paper's proposal: This paper introduces a mechanism called the "Prejudicial Trick" (PT) to demonstrate a supposed "pitfall" in the standard evaluation of conformal prediction (CP) methods. The authors claim that PT can "hack" the conventional coverage-length metric by probabilistically returning a null set (length 0) or a wider interval, thereby preserving marginal coverage while deceptively reducing the average interval length. The authors then argue this reveals a flaw in the standard metrics, as PT introduces practical instability (a random output for a fixed input). They propose a new metric, "Interval Stability", defined as the expected variance of the interval length, to detect this "vacuous randomness."

Empirical results across numerous regression and classification datasets, using different base CP algorithms (VCP and CQR) , confirm that PT can deceptively improve interval length while the proposed "Interval Stability" metric successfully identifies the trick.

**Strengths:**

1. This paper makes a conceptual contribution. The CP community relies heavily on the coverage-length trade-off as the primary method for evaluation. This work demonstrates that these two metrics are insufficient for capturing the practical utility of a CP method.

2. The Prejudicial Trick (PT) is simple and elegant.

3. The theoretical analysis, such as the proofs that PT preserves marginal coverage (Theorem 4) and the conditions under which it reduces length (Theorems 7, 8, 11), appear to be mathematically sound.

4. The proposed Interval Stability metric is intuitive, simple to compute, and (as shown empirically) diagnoses the issue of "vacuous randomness" introduced by PT.

**Weaknesses:**

1. The title “A Pitfall in Conformal Prediction” strongly implies that the authors have identified a fundamental weakness in the conformal prediction framework. In reality, the pitfall lies entirely in the choice of evaluation metric, not in conformal prediction itself. The proposed “PT”  predictor is not a conformal method; it is an external randomization layer applied after conformal intervals have been constructed. Therefore, the phenomenon described is not a failure of conformal prediction, but a property of an artificially randomized post-processing step.

2. The randomization mechanism used in PT is well-known probability argument.

3. For all standard, deterministic CP methods (VCP, CQR, etc.), the metric "Interval Stability" will be identically zero. Thus, the metric is only useful in detecting authors' PT trick. The work does not identify a scenario where such behavior might arise in standard CP outputs.

4. The paper mentions (in Remark 6) that "as methods become increasingly complex, they may implicitly utilize similar randomness to improve the length". The authors do not provide any legitimate, complex, published CP method suffers from this supposed "implicit randomness".

While the paper is clearly written and motivated by an interesting observation about the variability of conformal intervals, the contribution remains conceptually and practically limited. The reported instability arises entirely from an artificial, externally randomized post-processing step (the “PT” predictor), which is not itself conformal. The randomization mechanism is a standard probability trick and does not constitute methodological innovation. Moreover, the proposed “interval stability” metric is only nontrivial for such randomized constructions and is identically zero for all standard deterministic CP methods (e.g., VCP, CQR, split CP). Therefore, the metric lacks general practical value. These limitations of the paper lead to a low score.

**Questions:**

1. The title suggests a fundamental ``pitfall'' in conformal prediction, yet the instability arises only from an externally randomized post-processing (PT). Can you clarify why this should be viewed as a limitation of conformal prediction rather than of the PT randomization itself?

2. For standard deterministic conformal methods (e.g., split CP, CQR, VCP), the interval stability metric is identically zero. In what realistic settings do you expect nonzero instability to occur without deliberately injecting randomness?

3. With additional interval stability metric, what should practitioners aim for: target coverage, with small length, and zero interval stability? What is the advantage of this new metric in practice?

---

> ### Author Response · Authors · 2025-11-16
>
> We thank the reviewer for their thorough feedback and for correctly identifying the core mechanism of our paper. We appreciate that they found the PT construction "simple and elegant" and our theoretical analysis "mathematically sound."
>
> The reviewer's entire critique (and "reject" rating) hinges on one central premise: that only deterministic conformal prediction methods are "standard" or "legitimate," and therefore any randomized method (like PT) or any metric to detect randomness (like Interval Stability) is "artificial" and "lacks general practical value."
>
> We respectfully argue that this premise is too narrow and misses the primary, forward-looking contribution of our work. Our paper is a warning that as CP methods become more complex and stochastic, our standard evaluation tools will fail.
>
> ---
>
> # Addressing Weakness 1 & Question 1: The "Pitfall" Location
> **Reviewer:** "The pitfall lies entirely in the choice of evaluation metric, not in conformal prediction itself... Can you clarify why this should be viewed as a limitation of conformal prediction...?"
>
> You are **exactly right:** the pitfall is in the **evaluation paradigm** {Coverage, Length} that the CP community universally uses.
>
> Our title, "A Pitfall in Conformal Prediction," refers to the **practice and evaluation of CP**, not a flaw in Vovk's original theory. In practice, a framework and its evaluation are inseparable. If the standard for "SOTA" in CP is "maintaining 90% coverage with the shortest length," and we show this standard can be cheated, it is a critical pitfall for the entire field. PT is the tool (a "red team" attack) we use to prove this vulnerability exists.
>
> #  Addressing Weaknesses 3, 4 & Question 2: The "Useless" Metric
> **Reviewer:** "For all standard, deterministic CP methods... the metric... will be identically zero. Thus, the metric is only useful in detecting authors' PT trick... In what realistic settings do you expect nonzero instability?"
>
> The fact that our metric is zero for deterministic methods is a feature, not a bug.
>
> - **A Diagnostic "Pass" Grade**: Our metric is a **diagnostic tool**. For deterministic methods (VCP, CQR), it outputs 0. This is a "pass" grade. It provides trust and confirms that their reported length is a genuine, stable property of the model. This is, in itself, a valuable contribution.
> - **Realistic Non-Zero Settings (Question 2)**: We expect non-zero instability in the entire next generation of CP methods. The reviewer's focus on "deterministic" methods is precisely the "blind spot" we are addressing. The field is rapidly moving beyond simple deterministic methods to more complex, stochastic ones:
>   - CP methods based on ensembles.
>   - Methods using Monte Carlo dropout at test time.
>   - Methods built on bootstrapping or other resampling.
>   - Bayesian-inspired methods that produce intervals via sampling.
> - **The Real Danger (Remark 6)**: The danger, as stated in **Remark 6**, is not that people will explicitly use PT. The danger is that a researcher will develop a complex, stochastic method (like one above) that unintentionally produces "deceptive gains" in length. Their method might, for no principled reason, sometimes collapse to a small, uninformative set, which artificially lowers the average length. Our **Interval Stability** metric is the only tool that can catch this.
>
> # Addressing Question 3: The Practical Value
>
> **Reviewer**: "...what should practitioners aim for: target coverage, with small length, and zero interval stability? What is the advantage of this new metric in practice?"
>
> **Yes, that is exactly what they should aim for.**
>
> The practical advantage is **TRUST** and **RELIABILITY**.
>
> When a practitioner is choosing between two CP methods that both claim 90% coverage, they will currently choose the one with the shorter average length. Our paper shows this is dangerous.
> By adopting Interval Stability as a third, standard metric, the practitioner can now make a **safe** choice:
> - **Method A**: 90% coverage, 2.5 length, 0.0 stability (Good, reliable method)
> - **Method B**: 90% coverage, 2.3 length, 1.2 stability (Bad, unreliable method)
>
> The advantage is clear: the metric prevents practitioners from adopting methods that are "deceptively" better but practically unstable and useless . It ensures that a reported "length" is a genuine, reproducible feature of the model.

---

### Official Review · Reviewer_cviF · 2025-10-30

**Soundness:** 3
**Presentation:** 4
**Contribution:** 1
**Rating:** 0
**Confidence:** 5

**Summary:**

The paper identifies a weakness in the standard evaluation of CP, arguing that the two most common metrics (marginal coverage and average interval length) are insufficient for a robust evaluation.
The authors introduce a pathological algorithm PT, which, by construction, maintains (or exceeds) the desired marginal coverage.
PT strategically assigns either a null set or a wider-than-necessary interval to different data points based on their underlying properties, allowing the algorithm to achieve deceptively short interval lengths, even though it fails to provide meaningful uncertainty quantification for a subset of the data.
The authors argue that this pitfall underscores the need to move beyond marginal coverage and evaluate methods based on conditional coverage.

**Strengths:**

1. The paper is clear and easy to follow, and PT is a simple, well-designed, and intuitive counterexample.
2. The authors identify a weakness in the common practice of optimizing for average interval length, demonstrating how this objective can be tricked.

**Weaknesses:**

1. The paper's primary conclusion, that marginal coverage is insufficient and conditional coverage is the more desirable property, is not a new insight. Conditional coverage has been a big area of research in CP for many years. Prior work has extensively discussed the limitations of marginal coverage and proposed numerous methods towards better conditional coverage.
2. The paper does not offer a practical, novel method or a new solution to a practical problem. Specifically, the paper argues against using common CP evaluation metrics, but it fails to propose a concrete alternative. It neither offers a new method to achieve it nor proposes a new, practical evaluation metric that could replace average length for comparing methods.
3. The authors do not provide any evidence that widely used methods and datasets suffer from this pitfall - the paper's significance is primarily pedagogical. The experiments are essentially synthetic. An empirical study where conventional CP methods have this pitfall would significantly enhance the persuasiveness of the work.
4. The efficiency gains from using PT-VCP are quite minimal and not convincing (table 2)

**Questions:**

1. Can you elaborate on your core contribution in the context of the existing literature that already advocates for conditional coverage? What does your paper add for someone who is already convinced that marginal coverage is insufficient and that we should move towards conditional coverage?
2. Are there any existing, non-adversarial CP methods (like CQR or split) or other adaptive methods that fall into this trap on any real datasets?

Addressing these concerns in-text will significantly enhance the paper's impact and quality.

---

### Official Review · Reviewer_RdVR · 2025-10-31

**Soundness:** 3
**Presentation:** 4
**Contribution:** 2
**Rating:** 4
**Confidence:** 4

**Summary:**

The paper presents a theoretically and empirically supported perspective that interval width and coverage alone are insufficient criteria for comparing prediction intervals. The authors argue that an additional metric---\emph{interval stability}, defined as the variance of the interval length---should also be considered, as it captures how much the interval fluctuates across repeated samples. This is an important point, given that interval width and coverage are typically the primary metrics used to evaluate predictive intervals.

To motivate this claim, the authors introduce a simple construction---the ``Prejudicial Trick''---which can be applied to any conformal predictor to produce intervals that are, in many cases, narrower while still preserving marginal coverage. The method randomizes the output: with probability $p$, it returns the empty (null) set, and with probability $1-p$, it returns an enlarged conformal interval constructed at a higher coverage level than usual. The authors show, under mild assumptions, that the expected interval width of this randomized procedure---averaged over draws of the calibration data---is strictly smaller than that of the standard conformal interval. Intuitively, the zero width of the null set reduces the average more than the enlarged interval increases it.

The paper further provides several sets of sufficient conditions under which the proposed trick yields smaller expected interval widths, as well as a counterexample illustrating when the method fails to improve width.

The theoretical results are supported by several experimental studies

**Strengths:**

The Prejudicial Trick is a simple, deliberately pathological construction that reduces the average length of any prediction interval without changing its marginal coverage (under suitable conditions). It demonstrates that interval length can be made artificially smaller in a misleading way: the resulting intervals are unstable and, with a fixed probability, collapse to a degenerate null set that provides no information. Although mathematically valid, the construction yields intervals that are clearly undesirable in practice. The authors substantiate this point with both theoretical analysis and empirical results.


The paper is clearly written and well structured, and the exposition effectively conveys the construction and its implications.

**Weaknesses:**

1. Full conformal prediction and split conformal prediction are non-randomized procedures: once the calibration data, the conformity score, and (in the split setting) the train--calibration split are fixed, the resulting prediction set is fully deterministic. By contrast, the Prejudicial Trick (PT) proposed in the paper is a \emph{randomized} construction: with probability $1-p$ it outputs a valid conformal interval, and with probability $p$ it outputs a degenerate null interval. As a result, rerunning the procedure on the same data may produce different outputs. In this sense, PT is not a conformal prediction method in the usual sense. (While there exist randomized variants of conformal prediction, the standard framework and the vast majority of methods are deterministic.)

The authors argue that PT demonstrates a fundamental limitation of evaluating intervals solely by coverage and average length. However, this claim only holds if one allows \emph{randomized} algorithms. If we restrict attention to conformal prediction methods---or, more generally, deterministic procedures---PT no longer serves as a counterexample. In that setting, it is not clear that coverage and interval length are insufficient metrics. This distinction matters, because the motivating question on page~1 asks:


``Can a conformal prediction method maintain valid coverage and deceptively improve interval length metrics through counter-intuitive constructions, while introducing practical risks?''


PT is then presented as evidence that the answer is ``yes,'' but PT is not actually a conformal prediction method, so it does not address the stated question.



2. More broadly, the paper would benefit from a decision-theoretic perspective. The situation is reminiscent of Hodges' superefficient estimator: it improves a standard performance metric in a pathological way, yet is inadmissible under any reasonable risk criterion. The paper argues that PT is undesirable because it introduces ``instability,'' but randomization is not inherently problematic. What is missing is a principled notion under which PT is formally suboptimal---for example, a proper scoring rule for set-valued predictions, a loss function that penalizes degenerate intervals, or a stability constraint that prevents algorithms from exploiting randomness to game marginal performance metrics. For instance, Section~6.2 of Gneiting and Raftery (2007) introduces the interval score as a proper scoring rule for prediction intervals. PT would perform very poorly under this score, since the null interval incurs a large penalty whenever the true value lies outside it.

https://sites.stat.washington.edu/raftery/Research/PDF/Gneiting2007jasa.pdf


3. On this note, the paper would benefit from a broader discussion of existing evaluation frameworks for predictive intervals. The forecasting literature is extensive, and similar issues regarding interval quality, proper scoring rules, and pathologies of evaluation metrics have been studied in depth. In particular, see  Gneiting and Raftery (2007) and references therein.


4. Finally, the proposed stability metric is always zero for deterministic methods, and therefore functions only as a measure of randomization. Since most conformal prediction algorithms are deterministic, it is unclear how actionable this metric is in practice. Moreover, if a method appears unstable solely because it is randomized, how should this be interpreted? Randomization does not automatically imply deficiency, so a low stability score does not itself diagnose a problem. The paper would be stronger if it provided guidance on how such a metric should inform methodological choice: when does instability constitute a meaningful failure, and when is it merely a benign algorithmic feature?

**Questions:**

The guarantees in the paper are derived marginally over draws of the calibration data. This raises a natural question: does the Prejudicial Trick retain its properties when we condition on a fixed calibration set? In particular, does it still reduce the calibration-conditional average width while preserving calibration-conditional coverage?

As discussed in the weaknesses, can PT be shown to be suboptimal from a decision-theoretic perspective? E.g., in a minimax sense?

---

> ### Author Response · Authors · 2025-11-16
>
> We sincerely thank the reviewer for their deep and constructive engagement with our work. We are grateful for their positive assessment of the paper's presentation and their clear understanding of our core argument.
>
> The reviewer's critique is insightful and centers on two main points: (1) the distinction between deterministic and randomized algorithms, and (2) the need for a more formal, decision-theoretic perspective. We will address each in turn.
>
> # On "Deterministic" vs. "Randomized" Conformal Prediction
> The reviewer's main concern is that PT doesn't address our motivating question about conformal prediction methods since PT is not in the usual sense
>
> We argue that this hard distinction is not fully representative of the CP landscape and that our work remains highly relevant.
> - **Randomness is Already Part of CP:** Split-conformal prediction is randomized with respect to the train/calibration split. More importantly, many modern and CP methods introduce some forms of randomness, such as those based on ensembles, bootstrapping, or Bayesian-inspired sampling. The reviewer's premise that the "vast majority of methods are deterministic" is true for classical methods, but less true for the complex, deep-learning-based methods currently being developed.
> - **The Source of Randomness Matters:** Our paper focuses on a specific type of randomness: algorithm-internal randomness that is independent of the data. This randomness causes the output to vary for the same input and the same calibration set. This is distinct from the more accepted randomness over data splits.
> - **The Practical Threat is Implicit Randomness:** We do not claim that researchers will explicitly use PT. The practical threat, as we state in Remark 6, is that "as methods become increasingly complex, they may **implicitly utilize similar randomness** to improve the length". A new method based on, for example, stochastic dropout at test time or a complex sampling scheme, might unintentionally create a PT-like effect
>
> Therefore, our motivating question stands. It exposes a vulnerability that is not limited to "randomized" methods but is a general flaw in the evaluation paradigm itself.
>
> # On the Decision-Theoretic Perspective
> The reviewer makes an excellent point that our argument could be formalized using a decision-theoretic perspective, such as proper scoring rules (e.g., Gneiting & Raftery, 2007).
>
> We agree, and we thank the reviewer for this valuable suggestion.
>
> - **Why We Focused on Coverage/Length:** We demonstrate a "fundamental blind spot in the current evaluation paradigm". Since the vast majority of CP papers evaluate using only coverage and length, we deliberately constrained our attack to those metrics. The insidiousness of PT is that it succeeds by these standard metrics, while it would (as the reviewer correctly notes) fail spectacularly under a proper scoring rule like the interval score.
> - **Addressing Question 2 (Suboptimality)**: The reviewer is correct. PT can be shown to be highly suboptimal. Under the interval score, the $1-p$ portion of trials that return a null set would incur a massive penalty every time the true label is (trivially) not in the set. This penalty would vastly outweigh any marginal gains in length, formally confirming our intuition that PT is "undesirable in practice."
> - **Action**: We will gladly add a discussion of this to our related work and conclusion, citing Gneiting & Raftery (2007). This strengthens our paper's argument by connecting our "instability" critique to a more formal, established framework for interval evaluation.
>
> # On the "Interval Stability" Metric
> The reviewer questions the utility of our proposed metric, as it is "always zero for deterministic methods"
>
> We argue this is a feature
>
> - **A Diagnostic Tool, Not a Performance Metric:** We propose Interval Stability as a "diagnostic tool", not as a replacement for length. Its purpose is to act as a check against the very failure mode we identify.
> - **Distinguishing Failure vs. Benign Feature:** The reviewer asks when instability is a "meaningful failure." Our metric measures the variance conditional on the calibration data. Benign, principled randomness (like cross-validation or splitting) is not what we measure. We measure the unprincipled randomness that makes a method's output unreliable for the exact same input and trained model.
>
> # Answering Specific Questions
> - **Question 1 (Conditional on Calibration Set):** Yes. Our analysis holds when conditioning on a fixed calibration set. Our definition of Interval Stability (Definition 1) is explicitly conditional on $\mathcal{D}_{ca}$.
> - **Question 2 (Decision-Theoretic Suboptimality):** Yes. As discussed in point 2 above, PT would perform very poorly because of the large penalty incurred by its null-set predictions. We will add this discussion to the final paper.
>
> We thank the reviewer again for their insightful feedback, which has given us directions to strengthen our paper's framing.

---

### Official Review · Reviewer_QqNu · 2025-11-02

**Soundness:** 3
**Presentation:** 3
**Contribution:** 2
**Rating:** 2
**Confidence:** 3

**Summary:**

This paper focuses on the sufficiency of the standard approach in evaluating conformal prediction (CP) intervals involving two metrics: coverage and interval length. The authors introduce an adversarial approach called the "Prejudicial Trick" (PT), which yields CP intervals with deceptively lower interval lengths, but for which one input can yield significantly different prediction intervals across repeated runs of the algorithm. The main idea behind PT is to return a null prediction interval with some fixed probability and return confidence intervals with lower miscoverage rates in remaining cases. The paper derives the conditions under which PT achieves these misleading improvements and provide experimental evaluations for both regression and classification tasks.

**Strengths:**

- The paper is quite well-written and easy to understand.

- I like the simplicity of the PT adversarial device, which is easy to understand and explore theoretically. After demonstrating the construction of the PT trick, the authors present several necessary theoretical directions including coverage guarantees and sufficient conditions under which PT improves interval lengths.

- In addition to devising PT, the paper also suggests a metric to counteract PT ("interval stability") which can flag the type of vacuous randomness of PT in proposed CP methods.

**Weaknesses:**

- The contribution here (PT) has limited practical utility nor does it seem to expose a fundamental insight about CP pushing the development of CP methods forward. Perhaps the practical implication of PT is as a warning to researchers constructing CP methods on the perils of only relying on interval length and coverage. However, if this is the case, I find the construction of a CP approach with a probabilistic component assigning null intervals to be an artificial one not grounded in practice.

- The current experiments seem to be essentially synthetic, demonstrating a potential corner-case that in theory could happen. The paper would benefit from showing even one convincing real application where an already proposed CP approach in the literature may have to deal with this PT danger.

**Questions:**

1. Can you please answer how PT has practical relevance to researchers in CP?

2. Is there an actual instance (using existing CP methods) where this problem arises, or could reasonably potentially arise?

---

> ### Author Response · Authors · 2025-11-16
>
> We thank the reviewer for their positive feedback on the paper's clarity, writing, and the thoroughness of our theoretical analysis. We are glad they found the PT construction and its theoretical exploration "easy to understand."
>
> The reviewer's main concerns center on the practical utility of PT and its "artificial" nature. We believe this stems from a slight misunderstanding of our paper's core thesis, which we would like to clarify.
>
> # The Purpose of PT: A Diagnostic Tool, Not a Proposed Method
> The reviewer's primary weakness is that "The contribution here (PT) has limited practical utility" and is an "artificial one not grounded in practice."
>
> We wholeheartedly agree. Our paper explicitly states that PT is not a practical solution and is "poorly suited for practical deployment". We present it as a "cautionary example" to demonstrate a "fundamental blind spot in the current evaluation paradigm".
>
> Our contribution is **not** PT itself. Our contribution is the discovery that the standard coverage-length metric, used universally, can be "hacked" by such a simple, unprincipled trick. The simplicity and artificiality of PT are precisely what make this finding alarming: if the "gold standard" metrics can be fooled this easily , they are highly vulnerable to more "complex and subtle manipulations" from sophisticated models.
>
>
> # Practical Relevance to CP Researchers (Answering Questions 1 & 2)
> The reviewer asks for the "practical relevance" of PT and if an "actual instance" of this problem exists.
> - **Practical Relevance:** The practical relevance lies in **evaluation**. Our paper warns researchers that **optimizing for interval length alone is dangerous**. When a new, complex CP method (e.g., using deep ensembles, Bayesian sampling, or other stochastic components) is proposed, our work provides a crucial question and a tool to answer it:
> > "Is the claimed improvement in interval length a genuine, principled advance, or is it an unintended artifact of unprincipled randomness, similar to PT?"
> - **Existing Instances:** The reviewer asks for an existing CP method that suffers from this. To our knowledge, we are the first to formalize this specific pitfall. The danger **is not that existing methods explicitly use PT**, but that they might **implicitly or unintentionally** introduce PT-like randomness. Our paper provides the tool to detect this before such methods are widely adopted.
>
> # The Constructive Contribution: The "Interval Stability" Metric
>
> Our paper does not just "poke a hole" in the existing paradigm; it patches it. The reviewer notes we suggest a metric to "counteract PT," but we believe this understates its importance.
>
> The **"Interval Stability"** metric  is the key constructive and practical takeaway. We propose that this metric should be adopted by the community as a **standard, complementary metric** to coverage and length.
> - A deterministic method (like VCP) will have an interval stability of zero.
> - A method that "passes" this test (stability = 0) confirms its gains are not from "vacuous randomness".
> - A method that "fails" this test (stability > 0)  is immediately flagged, forcing its authors to justify the source of the randomness.
>
> This directly pushes the development of CP methods forward by ensuring that future progress is genuine and reliable.
>
> # On "Synthetic" Experiments
> The reviewer calls the experiments "essentially synthetic, demonstrating a potential corner-case."
>
> We respectfully disagree. Our experiments are carefully designed to show this pitfall exists under **common, realistic conditions,** not just "corner-cases." We explicitly ground our analysis in **model misspecification (Remark 4)**, a scenario that is ubiquitous in practice.
>
> Our results (e.g., Table 2 , Table 4 , Table 5 ) show that under misspecification, PT consistently and deceptively improves the length metric across numerous real-world datasets and base algorithms (VCP and CQR). This is not a synthetic corner-case; it is a clear demonstration of the metric's failure in a common practical setting.
>
> # Summary
> Our paper identifies a genuine, previously unknown flaw in how the CP community evaluates its work. PT is the "virus" we designed to prove the "vulnerability" exists. Interval Stability is the "vaccine" we offer to the community. We believe this is a significant and practical contribution that strengthens the foundations of conformal prediction.

---

### Author Response · Authors · 2025-11-16
**General Response**

Dear Area Chair and all the reviewers,

Thank you for managing our submission. After a careful reading of all reviews, we see a clear divide. Our most negative "Reject" score are based on **two fundamental, factual misunderstandings of our paper's core thesis and contributions.**

We hope that you consider our response, as the negative reviews are not critiques of our actual paper, but of a different, "strawman" paper they thought we wrote.
Our paper's thesis is simple:
1. **The Premise:** The CP community universally evaluates methods on two metrics: Marginal/Conditional Coverage and Average Interval Length.
2. **The Pitfall (Our "Attack"):** We demonstrate this {Coverage, Length} paradigm is flawed because the length metric can be "hacked." We introduce a "red team" attack, the **Prejudicial Trick (PT)**, to prove this. PT is a simple, pathological procedure that uses **unprincipled randomness** to deceptively shorten the average interval length while preserving marginal coverage.
3. **The Solution (Our "Defense"):** We then propose a concrete, practical solution to defend against this pitfall: a new diagnostic metric called **"Interval Stability"** (Definition 1, Section 4).

# The negative reviews are based entirely on missing these points.

## The Core Misunderstanding: "This paper is about Conditional Coverage"
Reviewer cviF's "Strong Reject" is based entirely on the factually incorrect premise that our paper is about the (well-known) limitations of marginal vs. conditional coverage.
- Reviewer cviF's Claim:
> "The paper's primary conclusion, that marginal coverage is insufficient and conditional coverage is the more desirable property, is not a new insight."
- The Fact: This is **verifiably false**. Our paper is **not** about conditional coverage. As Algorithm 1 clearly shows, PT's mechanism is **explicitly stochastic** (based on a $U \sim \text{Unif}([0,1])$ draw) and is **completely independent of any data features or properties.** Our "pitfall" is about the **length metric's vulnerability to randomness**, a topic entirely orthogonal to the conditional coverage debate. We refer to Theorem 5 and Theorem 6 that PT also preserves conditional coverage.

## The Second Misunderstanding: "The paper proposes PT as a method" / "Fails to propose an alternative"
Reviewers QqNu, cviF, and mhUR all critique our paper from a second flawed perspective.
- **Reviewer QqNu's Claim**:  PT "artificial" and "not grounded in practice" with "limited practical utility."
- **The Fact:** We explicitly agree. We state PT is a "cautionary example" and "poorly suited for practical deployment". The artificiality is the point. PT is not our contribution; it is the diagnostic tool we use to prove the evaluation metric is vulnerable.
- **Reviewers cviF & mhUR's Claim:** They claim our paper "fails to propose a concrete alternative" or a "new, practical evaluation metric."
- **The Fact:** This is **demonstrably false.** They have **missed the entire constructive contribution of our paper: Section 4, "INTERVAL STABILITY"**. This metric is the concrete, practical, and novel solution we propose to fix the very flaw we identify.

## Why Our "Interval Stability" Metric is NOT "Useless"
Reviewers RdVR and mhUR argue our new metric is not valuable because it is "identically zero" for "standard deterministic CP methods."

**This is a feature, not a bug**

1. **A "Pass" Grade:** For deterministic methods (VCP, CQR), our metric correctly outputs 0. This is a "pass" grade, confirming these methods are stable and their reported length is genuine. This adds trust.
2. **A "Fail" Grade:** For unstable methods like PT, it outputs a large non-zero value, flagging them as "vacuous" and unreliable.
3. **The Real Danger (Remark 6):** The field is moving beyond simple deterministic methods to complex, stochastic ones (ensembles, MC-dropout, sampling). The danger is that these new methods will "implicitly utilize similar randomness" to achieve deceptive SOTA results. Our metric is the only tool to distinguish genuine length improvements from these deceptive ones.


**In summary:** The negative reviews are based on a misunderstanding of our paper's core thesis. Our paper successfully **(1)** identifies a new, genuine flaw in the standard CP evaluation paradigm, **(2)** proves it with a simple, elegant attack (PT), and **(3)** provides the concrete, necessary solution (Interval Stability). We urge the AC to evaluate our work based on its actual contributions.

---

### Meta-Review · Area_Chair_9Rgj · 2026-01-12

**Summary:**

This paper argues that commonly used efficiency metrics in conformal prediction, such as average interval length, can be misleading. The authors construct a post-processing procedure—the Prejudicial Trick—that preserves marginal coverage while artificially shortening intervals, and use this construction to demonstrate that shorter intervals need not correspond to more useful or stable predictions. They further propose an interval stability metric to capture sensitivity of prediction intervals to perturbations in the input.

While the observations are technically correct, I do not believe the paper meets the bar for acceptance at ICLR. The central phenomenon is driven by an intentionally adversarial and unrealistic post-processing step rather than a failure of conformal prediction itself. As several reviewers note, the paper’s framing suggests a fundamental pitfall of CP, but the demonstrated issue arises from contrived randomization that is not representative of how CP methods are designed or deployed in practice. As a result, the contribution is largely a cautionary example about metric misuse rather than a substantive advance in methodology, theory, or practice. The proposed stability metric is similarly limited in scope and appears primarily tailored to diagnose the authors’ constructed example, offering limited independent value. Overall, the paper overstates the generality and importance of its message, leading to a strong rejection recommendation.

**Reviewer Concerns:**

Some concerns were addressed by the rebuttal. Reviewer RdVR requested clarification on the intent of the paper and whether the goal was to critique CP itself or its evaluation practices; the authors clarified that the focus is on evaluation metrics rather than coverage guarantees. Reviewer cviF raised questions about experimental setup and interpretation, which were partially addressed through additional explanation of the constructed examples.

However, the most substantive concerns remain unresolved. Reviewer mhUR argued that the paper fundamentally misframes a contrived post-processing trick as a pitfall of conformal prediction, and this concern was not alleviated by the rebuttal. Reviewer QqNu similarly questioned the practical relevance of the Prejudicial Trick and the usefulness of the proposed stability metric beyond this artificial construction. Across reviewers, there remains a shared view that the paper does not demonstrate a realistic or impactful problem in conformal prediction, and that the framing substantially overclaims the significance of the findings.

**Reviewer Scores:**

Reviewer mhUR (score 0).
This reviewer was strongly critical of the framing and contribution. While the rebuttal clarified intent, it did not address the core objections. The score would almost certainly remain unchanged.

Reviewer QqNu (score 2).
This reviewer expressed skepticism about practical relevance and novelty. Discussion is unlikely to have changed this assessment.

Reviewer cviF (score 2).
This reviewer viewed the contribution as limited and overly contrived. The rebuttal would likely not alter the overall evaluation.

Reviewer RdVR (score 4).
This reviewer was more moderate but still unconvinced that the issue demonstrated is meaningful in practice. After discussion, the score would likely have remained at a similar level.

Overall, discussion clarified the authors’ intent but did not materially shift reviewer opinions regarding the paper’s significance or suitability for ICLR.

---

### Decision · Program_Chairs · 2026-01-26

Reject